# *In-silico* genome wide analysis of Mitogen activated protein kinase kinase kinase gene family in *C. sinensis*

**Abhirup Paul**[1☯], **Anurag P. Srivastava**[2☯], **Shreya Subrahmanya**[3], **Guoxin Shen**[4☯]*, **Neelam Mishra**[3]*

**1** Department of Biochemistry, REVA University, Bangalore, Karnataka, India, **2** Department of Life Sciences, Garden City University, Bangalore, Karnataka, India, **3** Department of Botany, St. Joseph's College Autonomous, Bangalore, Karnataka, India, **4** Sericultural Research Institute, Zhejiang Academy of Agricultural Sciences, Hangzhou, China

☯ These authors contributed equally to this work.
* guoxin.shen@ttu.edu (GS); neelamiitkgp@gmail.com, neelammishra@sjc.ac.in (NM)

**Data Availability Statement:** All relevant data are within the manuscript and its Supporting information files.

## Abstract

Mitogen activated protein kinase kinase kinase (MAPKKK) form the upstream component of MAPK cascade. It is well characterized in several plants such as Arabidopsis and rice however the knowledge about MAPKKKs in tea plant is largely unknown. In the present study, MAPKKK genes of tea were obtained through a genome wide search using *Arabidopsis thaliana* as the reference genome. Among 59 candidate MAPKKK genes in tea, 17 genes were MEKK-like, 31 genes were Raf-like and 11 genes were ZIK- like. Additionally, phylogenetic relationships were established along with structural analysis, which includes gene structure, its location as well as conserved motifs, cis-acting regulatory elements and functional domain signatures that were systematically examined. Also, on the basis of one orthologous gene found between tea and Arabidopsis, functional interaction was carried out in *C. sinensis* based on an Arabidopsis association model. The expressional profiles indicated major involvement of MAPKKK genes from tea in response to various abiotic stress factors. Taken together, this study provides the targets for additional inclusive identification, functional study, and provides comprehensive knowledge for a better understanding of the MAPKKK cascade regulatory network in *C. sinensis*.

## Introduction

Mitogen-activated protein kinase (MAPK) cascades are universal signal transduction modules existing in eukaryotes, including yeasts, animals and plants. MAPKKKs (Mitogen activated protein kinase kinase kinase), which form the upstream component of three tier kinase module are usually activated by G-proteins (Guanine nucleotide binding protein) but sometimes activation is also done via an upstream MAP4K [1]. MAPKKKs are the first component of this phosphorelay cascade, which phosphorylates two serine/threonine residues in a conserved S/T-X$_{3-5}$-S/T (Serine/Threonine-X$_{3-5}$-Serine/Threonine) motif of the MKK (Mitogen activated

**Funding:** This project was supported by the Key Research and Development Program of Zhejiang Province (No. 2021C02002)?the Key Technologies R & D Program for Crop Breeding of Zhejiang Province (2021C02072-5), the Natural Science Foundation of China (31402140?31670303).

**Competing interests:** The authors have declared that no competing interests exist.

protein kinase kinase) activation loop. MKKs that are dual-specificity kinases, activate the downstream MAPK through TDY or TEY phosphorylation motif in the activation loop (T-loop) [2, 3]. The activated MAPK ultimately phosphorylates various downstream substrates, including transcription factors and other signalling components that regulate the expression of downstream genes [4]. MAPKKKs form the largest group among MAPK cascade, with 80 members in Arabidopsis, 75 members in rice, 74 members in maize and 89 members in tomato [5, 6]. This largest group is further subdivided into three smaller groups on the basis of sequence similarities 1) MEKK subfamily 2) Raf subfamily 3) ZIK subfamily [6, 7]. Compared to MAPKs and MAPKKs, the MAPKKKs have more members and greater variety in primary structures and domain composition [8]. Phylogenetic analysis of the MAPKKK genes in various species reveals the diversity in plants. Among the MAPKKKs, the Raf subfamily is the largest group and comprises of 46 members in maize, 43 in rice, 27 in grapevines, and 48 in Arabidopsis. It is followed by the MEKK subfamily, which is the second largest family and comprises of 22 members in maize, 22 in rice, 9 in grapevine, and 21 in Arabidopsis. The ZIK subfamily is the smallest among the three subfamilies and comprises of 6 members in maize, 10 in rice, 9 in grapevines, and 11 in Arabidopsis [5, 6, 9]. The MEKK subfamily comprises of a conserved kinase domain G(T/S)Px(W/Y/F)MAPEV [5]. The ZIK subfamily contains TPEF-MAPE(L/V)Y while the Raf subfamily has GTxx(W/Y)MAPE as their conserved domain signatures [5]. All the MAPKKK proteins have a kinase domain, and most of them have a serine/threonine protein kinase active site [10]. Structural domain analysis of MAPKKKs in Arabidopsis, rice and cucumber showed that most of the Raf proteins have a C-terminal kinase domain and a long N-terminal regulatory domain. In contrast, members of the ZIK group have the N-terminal kinase domain, while members of the MEKK group have a less conserved kinase domain that lies in either N or C-terminals or present in the central part of the protein [6, 9, 11]. MAPKKKs play a significant role in distinct biological and physiological processes, and they have potential that could be utilized for the development of stress-tolerant transgenic plants [12]. Two of the best studied Arabidopsis MAPKKKs are EDR1 (Enhanced disease resistance) and CTR1 (Constitutive triple response) which are known to participate in defense responses and ethylene signalling respectively [2, 13, 14].

*Camellia sinensis* more commonly known as tea is the second most consumed beverage in the world besides water. Tea plant is an important commercial crop potentially rich in variety of bioactive ingredients. Many genome wide studies of different gene families have been carried out in tea however, the MAPKKK genes and its role in stress response in tea plant have not been studied in detail. In the present study, the MAPKKK family of genes was thoroughly defined on the basis of *in-silico* genome-wide search in tea using *Arabidopsis thaliana* as the reference genome. Gene locations on scaffolds, their structures, the cis-regulatory elements and their evolutionary aspect were systematically studied. Further, we analysed the interaction networks of proteins based on orthologous genes in Arabidopsis. This study provides an insight on structural and functional aspect of Mitogen Activated Protein Kinase Kinase Kinase gene family in *C. sinensis* and also highlights the MAPK signalling cascade-mediated pathway of *C. sinensis*.

## Materials and methods

### Identification of MAPKKK gene family in tea

The predicted peptide sequences of tea were downloaded from the Tea Plant Information Archive (TPIA) database (http://tpia.teaplant.org/) [15]. To identify tea MAPKKK genes, a total of 415 previously known MAPKKK genes were retrieved from *Arabidopsis thaliana* (80), *Oryza sativa* (75), *Solanum lycopersicum* (71), *Solanum tuberosum* (81), *Capsicum annum* (60)

and *Coffee canephora* (48) using TAIR database (https://www.arabidopsis.org/) [16], Rice Genome Annotation Project database (http://rice.plantbiology.msu.edu/) [17] and Sol Genomics Network database (https://solgenomics.net/) [18], respectively. The retrieved Arabidopsis and rice MAPKKK sequences were used as query sequences to search against the tea plant proteome database using the BLASTp algorithm with an e value set to 1e-5 and identity percentage of 50% as threshold. The identified sequences were checked to remove any chances of redundancy. Further, the obtained genes were aligned by CLUSTALW (https://www.ebi.ac.uk/Tools/msa/clustalo/) [19] and uploaded to SMART (http://smart.embl-heidelberg.de/) [20] and Pfam web tool (https://pfam.xfam.org/) to confirm the existence of kinase domains. The physicochemical properties of the identified tea MAPKKK genes were predicted using Prot-Param tool incorporated in ExPASy database (https://expasy.org/) [21]. Subcellular localization of the peptides were predicted using the BaCelLo (Balanced subcellular localization predictor) (http://gpcr.biocomp.unibo.it/bacello/index.htm) [22] and TMHMM server v2.0 (http://www.cbs.dtu.dk/services/TMHMM/) [23] was employed to predict the presence of trans-membrane helices in tea MAPKKK peptide sequences.

## Estimation of $K_a/K_s$ ratios

$K_a$ and $K_s$ ratios were calculated using the SNAP v.2.1.1 online tool (https://www.hiv.lanl.gov/content/sequence/SNAP/SNAP.html) [24] to assess the synonymous and non-synonymous groups. The dN/dS values represent the selective pressure of duplicate genes and the dS values represent the time of divergence of duplication events.

## Multiple sequence alignment and phylogeny analysis

The tea MAPKKK protein sequences were subjected to multiple sequence alignment, using CLUSTALW (https://www.ebi.ac.uk/Tools/msa/clustalo/) [19] to check for conserved MAPKKK specific domains for each subfamily. Phylogenetic analyses were done separately for MEKK, Raf and ZIK sub-families, using the identified tea sequences, coupled with Arabidopsis, rice, tomato, potato, capsicum and coffee peptide sequences. The phylogenetic trees were constructed by the Neighbor-Joining algorithm of MEGA 7.0.14 [25] keeping all the parameters at default values. The consistencies of the obtained trees were assessed by the bootstrap method and replicate was set to 1000.

## Intron exon structures and conserved motifs

The intron exon distribution pattern for tea MEKK, Raf and ZIK peptide sequences were analysed and visualised using the Gene Structure Display Server v2.0 (http://gsds.cbi.pku.edu.cn/) [26]. The full-length peptide sequences were uploaded to MEME suite (http://meme-suite.org/) [27] in-order to identify the conserved motifs.

## Analysis of cis-regulatory elements

The promoter sequences of 2000 bp, which lies upstream of the translational start site of each of the tea MAPKKK genes were retrieved from the TPIA database. The PlantCARE database (http://bioinformatics.psb.ugent.be/webtools/plantcare/html/) [28, 29] was used for identifying and analysing the cis-acting regulatory elements in the promoter regions of the tea MAPKKK genes.

## Mapping of tea MAPKKK genes onto scaffolds and gene duplication

TPIA database has incomplete genome assembly information. As a result, the tea MAPKKK genes were mapped onto their respective scaffolds using MapGene2chromosome web v2 (MG2C) software tool (http://mg2c.iask.in/mg2c_v2.0/) [30]. The genes were mapped according to their scaffold positional information available in TPIA database, which includes scaffold IDs for each gene, scaffold dimensions and the starting and ending position of each gene on the scaffolds.

## GO ontology annotation and functional interaction network

All the identified tea MAPKKKs were searched individually to retrieve data of all the GO terms to which they have been annotated. The terms that appear for each searched gene, open up to AmiGO 2 server (AmiGO 2: Welcome (geneontology.org)), displaying detailed information for the respective GO terms. This data was then reconfirmed by searching the GO terms in QuickGO server (https://www.ebi.ac.uk/QuickGO/). The statistical significance of the GO terms were then checked using the TeaCoN database (https://teacon.wchoda.com) [31]. The obtained GO terms were divided into 3 main categories; Biological process (BP), Cellular component (CC) and Molecular function (MF). Furthermore, the orthologous groups search option was used to search for orthologous genes between tea and Arabidopsis. This includes all the *C. sinensis* gene accessions that are orthologous to other species including Arabidopsis (Orthologous Groups (shengxin.ren)). The orthologous gene obtained was then used to construct the network of functionally interacting orthologous gene between tea and Arabidopsis using STRING online tool ((https://string-db.org/) [32] with default parameters.

## Expression profiles of tea MAPKKK genes

The tissue specific expression profiles, which include expression levels in apical bud, flower, fruit, young leaf, mature leaf, old leaf, root, and stem were retrieved from TPIA database. Furthermore, gene expression data under different abiotic stress (cold, drought, salt) treatment as well as under methyl jasmonate (MeJA) treatment were retrieved from TPIA database. Graph-Pad Prism 8 (https://www.graphpad.com/scientific-software/prism/) was used to generate respective graphs for the gene expression data of MEKK, Raf and ZIK sub-families.

## Results

### Identification of MAPKKK gene family in *C. sinensis*

In order to identify the MAPKKK gene family in tea (*C. sinensis*), 415 known MAPKKK peptide sequences from *Arabidopsis thaliana* (80), *Oryza sativa* (75), *Solanum lycopersicum* (71), *Solanum tuberosum* (81), *Capsicum annum* (60) and *Coffee canephora* (48) were retrieved from their respective databases. To identify and categorize the MAPKKK genes in tea, BLASTp searches were conducted against the tea protein database, using the retrieved peptide sequences from Arabidopsis and rice as query sequences. For all BLASTp searches, e value and identity percentage were set to 1e-5 and 50% as threshold, respectively (S1-S3 Tables in S1 File). The identified tea peptides were again screened with a Hidden Markov Model (HMM) search to confirm the presence of serine/threonine-protein kinase-like domain (PF00069). The results yielded a total of 59 potential tea MAPKKK genes, which included 17 MEKK-like, 31 Raf-like and 11 ZIK-like genes and were incorporated into the final dataset.

The physicochemical properties of the identified tea MAPKKK protein sequences were evaluated using ExPASy ProtParam tool (Tables 1–3). The length and molecular weight of the 17 MEKK proteins ranged from 311 to 1191 amino acid residues and 34828.88 to 130956.46

**Table 1. Sequence characteristics and physicochemical properties of MAPKKKs belonging to MEKK subfamily in *C. sinensis*.**

| Gene ID | Locus position | Gene length (bp) | Protein length (aa) | Mol. Wt. (kDa) | pI value | No. of negative residues | No. of positive residues | GRAVY index | Instability index | Aliphatic index | Subcellular localization |
|---|---|---|---|---|---|---|---|---|---|---|---|
| TEA028357.1 | Scaffold856:196999-204246- | 7247 | 628 | 68667.76 | 5.60 | 77 | 67 | -0.380 | 58.36 | 76.85 | Nucleus |
| TEA025870.1 | Scaffold790:521648–539960+ | 18312 | 776 | 85271.15 | 6.76 | 94 | 92 | -0.379 | 45.58 | 81.08 | Nucleus |
| TEA016319.1 | Scaffold3144:371539-383072- | 11533 | 627 | 68238.67 | 9.50 | 53 | 71 | -0.535 | 50.61 | 68.23 | Nucleus |
| TEA008165.1 | Scaffold3102:729210-737275+ | 8065 | 1032 | 112285.36 | 9.04 | 84 | 102 | -0.423 | 53.62 | 72.95 | Nucleus |
| TEA027265.1 | Scaffold1289:966535–975893+ | 9358 | 939 | 101539.85 | 9.35 | 80 | 104 | -0.605 | 63.34 | 65.88 | Nucleus |
| TEA006319.1 | Scaffold2905:735285–744378+ | 9093 | 683 | 75479.57 | 9.32 | 62 | 78 | -0.505 | 67.84 | 72.55 | Chloroplast |
| TEA006473.1 | Scaffold1374:1527992-1535696- | 7704 | 710 | 78857.59 | 9.09 | 65 | 79 | -0.516 | 69.53 | 71.59 | Nucleus |
| TEA014429.1 | Scaffold41:2381991–2415462+ | 33471 | 1191 | 130956.46 | 6.13 | 145 | 128 | -0.350 | 45.47 | 89.93 | Chloroplast |
| TEA031711.1 | Scaffold5399:986467-998883- | 12416 | 562 | 62129.83 | 6.31 | 72 | 69 | -0.484 | 48.47 | 75.62 | Nucleus |
| TEA001470.1 | Scaffold558:920549–933450+ | 12901 | 789 | 87423.22 | 8.34 | 90 | 95 | -0.313 | 49.68 | 84.13 | Nucleus |
| TEA017119.1 | Scaffold5354:234291-239017- | 4726 | 506 | 56190.19 | 4.66 | 80 | 49 | -0.481 | 47.12 | 69.53 | Nucleus |
| TEA005306.1 | Scaffold2184:2097399–2125258+ | 27859 | 1097 | 121164.56 | 5.40 | 162 | 127 | -0.540 | 49.78 | 72.63 | Nucleus |
| TEA009902.1 | Scaffold438:521469-522821- | 1352 | 450 | 49874.37 | 4.58 | 65 | 34 | -0.060 | 44.64 | 91.60 | Chloroplast |
| TEA029598.1 | Scaffold944:301732–304329+ | 2597 | 423 | 46235.51 | 4.94 | 62 | 43 | -0.433 | 51.54 | 74.18 | Nucleus |
| TEA005122.1 | Scaffold1857:297670-298674- | 1004 | 334 | 36588.08 | 6.01 | 40 | 34 | -0.381 | 46.55 | 78.23 | Chloroplast |
| TEA028214.1 | Scaffold613:628014-629048+ | 1034 | 344 | 38088.50 | 6.33 | 44 | 41 | -0.322 | 45.18 | 79.36 | Nucleus |
| TEA031689.1 | Scaffold1549:309791-310726- | 935 | 311 | 34828.88 | 6.04 | 44 | 40 | -0.340 | 48.20 | 90.64 | Nucleus |

Locus position, gene length, protein length, molecular weight and pI value, no. of negative and positive residues, GRAVY index, instability index, aliphatic index and subcellular localizations were analysed.

kDa, respectively (Table 1). For the Raf proteins, it ranged from 305 to 1436 amino acid residues and 35012.57 to 159263.21 kDa (Table 2), and for the ZIK proteins, it ranged from 300 to 831 amino acid residues and 34181.96 to 94422.51 kDa (Table 3). The theoretical pI values ranged from 4.58 to 9.50 for MEKK, 4.88 to 9.61 for Raf and 5.14 to 6.33 for ZIK proteins, indicating that most of the MEKK and Raf proteins have a basic nature while the ZIK proteins are acidic in nature. The grand average of hydropathy (GRAVY index) in all the extracted MEKK, Raf and ZIK proteins were negative, ranging from -0.605 to -0.060, -0.661 to -0.182 and -0.582 to -0.350 respectively. This indicates that all the identified 59 tea MAPKKKs are hydrophilic in nature. 52 of the 59 putative tea MAPKKKs had instability index values above 40, while 6 Raf genes (TEA000933.1, TEA022171.1, TEA011280.1, TEA031223.1, TEA007232.1 and TEA013875.1) and 1 ZIK gene (TEA020112.1) had instability index values less than 40 (Tables 1–3). This signifies the unstable nature of most of the identified tea MAPKKKs. Subcellular localization predicted 48 genes being localized in the nucleus, 9 genes in chloroplast and 2 genes in cytoplasm (Tables 1–3). The presence of trans-membrane helices in the putative peptide sequences was also predicted and one of the ZIK gene (TEA027328.1) had one trans-membrane helix (S1-S3 Figs in S2 File).

## Phylogenetic analysis of tea MAPKKKs

A phylogenetic analysis of the putative tea MAPKKK genes was carried out to evaluate the evolutionary relationships. MEGA 7.0.14 was used to generate the phylogenetic trees, using the Neighbor-Joining (NJ) algorithm, at default parameters and 1000 bootstrap replicates. Three different phylogenetic trees were constructed for MEKK, Raf and ZIK proteins, comprising of the identified tea sequences and already known 415 MAPKKK sequences from Arabidopsis, rice, tomato, potato, capsicum and coffee. For MEKK, the NJ tree was generated using 17

**Table 2. Sequence characteristics and physicochemical properties of MAPKKKs belonging to Raf subfamily in *C. sinensis*.**

| Gene ID | Locus position | Gene length (bp) | Protein length (aa) | Mol. Wt. (kDa) | pI value | No. of negative residues | No. of positive residues | GRAVY index | Instability index | Aliphatic index | Subcellular localization |
|---|---|---|---|---|---|---|---|---|---|---|---|
| TEA001765.1 | Scaffold1670:382409–407933- | 25524 | 842 | 93193.15 | 5.86 | 107 | 92 | -0.248 | 46.33 | 89.69 | Nucleus |
| TEA002020.1 | Scaffold3595:726640–735244+ | 8604 | 896 | 99135.31 | 6.37 | 111 | 103 | -0.382 | 42.96 | 81.80 | Nucleus |
| TEA000256.1 | Scaffold3876:193108–215389+ | 22281 | 1086 | 119081.90 | 6.63 | 118 | 112 | -0.441 | 44.80 | 78.36 | Nucleus |
| TEA029086.1 | Scaffold106:745738–778269+ | 32531 | 919 | 101696.51 | 5.17 | 119 | 85 | -0.182 | 41.82 | 91.44 | Chloroplast |
| TEA022129.1 | Scaffold3036:784237–806418+ | 22181 | 940 | 104852.31 | 6.01 | 114 | 102 | -0.217 | 48.52 | 89.81 | Nucleus |
| TEA019143.1 | Scaffold1695:623368–630213+ | 6845 | 724 | 79987.28 | 7.68 | 86 | 87 | -0.609 | 41.33 | 70.98 | Nucleus |
| TEA028452.1 | Scaffold433:2415340–2426547+ | 11207 | 846 | 93141.05 | 6.10 | 107 | 93 | -0.523 | 46.31 | 70.89 | Nucleus |
| TEA016969.1 | Scaffold4925:453439–477111+ | 23672 | 1107 | 124661.25 | 8.46 | 145 | 153 | -0.506 | 46.58 | 77.06 | Nucleus |
| TEA013270.1 | Scaffold344:585774–608400+ | 22626 | 755 | 85320.07 | 5.83 | 104 | 84 | -0.374 | 53.97 | 80.97 | Nucleus |
| TEA026716.1 | Scaffold1930:511463–522712- | 11249 | 368 | 41783.40 | 5.63 | 52 | 44 | -0.487 | 46.26 | 73.89 | Nucleus |
| TEA028758.1 | Scaffold9739:380569–387825- | 7256 | 1213 | 135047.30 | 5.63 | 159 | 123 | -0.661 | 51.62 | 66.13 | Nucleus |
| TEA010804.1 | Scaffold35:1009695–1024064+ | 14369 | 305 | 35012.57 | 6.50 | 44 | 41 | -0.644 | 44.62 | 74.16 | Nucleus |
| TEA009451.1 | Scaffold1786:773656–783180- | 9524 | 1333 | 148469.64 | 4.88 | 193 | 127 | -0.547 | 45.20 | 72.24 | Nucleus |
| TEA021421.1 | Scaffold1504:1005366–1017261- | 11895 | 1331 | 147137.87 | 5.50 | 181 | 135 | -0.618 | 44.79 | 71.96 | Nucleus |
| TEA017670.1 | Scaffold1965:485241–501001+ | 15760 | 561 | 62970.57 | 5.67 | 78 | 62 | -0.385 | 49.24 | 89.63 | Nucleus |
| TEA019184.1 | Scaffold4191:163416–174412+ | 10996 | 601 | 67890.36 | 5.90 | 76 | 65 | -0.328 | 49.47 | 89.02 | Nucleus |
| TEA000933.1 | Scaffold397:63694–76812+ | 13118 | 407 | 45660.50 | 7.66 | 45 | 46 | -0.291 | 38.27 | 81.89 | Nucleus |
| TEA031230.1 | Scaffold2248:916505–925818+ | 9313 | 489 | 54655.73 | 9.20 | 56 | 67 | -0.311 | 43.48 | 87.32 | Chloroplast |
| TEA022171.1 | Scaffold382:2039496–2046332+ | 6836 | 404 | 44640.23 | 8.60 | 48 | 54 | -0.379 | 26.52 | 78.89 | Nucleus |
| TEA011280.1 | Scaffold3804:571784–578194+ | 6410 | 368 | 41126.17 | 7.52 | 48 | 49 | -0.312 | 25.58 | 82.91 | Nucleus |
| TEA031223.1 | Scaffold2248:107085–111327- | 4242 | 434 | 48234.42 | 6.42 | 57 | 55 | -0.280 | 26.84 | 84.22 | Nucleus |
| TEA007232.1 | Scaffold3038:2387807–2395630- | 7823 | 368 | 41118.03 | 7.02 | 48 | 48 | -0.424 | 35.65 | 77.34 | Nucleus |
| TEA016553.1 | Scaffold1761:1968012–1984037- | 16025 | 432 | 49062.23 | 8.45 | 65 | 69 | -0.513 | 43.90 | 83.06 | Nucleus |
| TEA033032.1 | Scaffold858:331961–346154- | 14193 | 415 | 46688.59 | 6.05 | 59 | 52 | -0.355 | 43.89 | 86.53 | Nucleus |
| TEA001764.1 | Scaffold619:1545624–1550286+ | 4662 | 351 | 39474.64 | 6.47 | 42 | 39 | -0.191 | 43.90 | 86.72 | Cytoplasm |
| TEA026000.1 | Scaffold3457:1062923-1073461- | 10538 | 1296 | 144765.39 | 5.40 | 177 | 122 | -0.507 | 42.22 | 73.72 | Nucleus |
| TEA033556.1 | Scaffold192:400250-403690- | 3440 | 541 | 61612.16 | 9.27 | 57 | 72 | -0.371 | 46.58 | 86.19 | Chloroplast |
| TEA013875.1 | Scaffold5449:126808–131150+ | 4342 | 341 | 39047.30 | 6.76 | 44 | 42 | -0.256 | 36.12 | 91.52 | Cytoplasm |
| TEA002722.1 | Scaffold1369:145416–156759+ | 11343 | 1436 | 159263.21 | 5.41 | 203 | 153 | -0.566 | 45.20 | 72.12 | Nucleus |
| TEA030052.1 | Scaffold319:1148438–1156900+ | 8462 | 1357 | 148194.52 | 5.00 | 166 | 110 | -0.444 | 50.17 | 73.96 | Nucleus |
| TEA008343.1 | Scaffold142:344598–352450+ | 7852 | 334 | 37992.05 | 9.61 | 35 | 46 | -0.257 | 46.78 | 86.17 | Nucleus |

Locus position, gene length, protein length, molecular weight and pI value, no. of negative and positive residues, GRAVY index, instability index, aliphatic index and subcellular localizations were analysed.

sequences from tea, 21 sequences from Arabidopsis, 22 sequences from rice, 17 sequences from tomato, 22 sequences from potato, 17 sequences from capsicum and 12 sequences from coffee (Fig 1A). The NJ tree was divided into 4 distinct clades, with an uniform distribution of genes in Clade A. Clade B consisted of only 6 capsicum genes while clade D had only 2 genes of potato. Clade C however, had a share of tomato and potato gene clusters. For Raf, the NJ tree was generated using 31 sequences from tea, 48 sequences from Arabidopsis, 43 sequences from rice, 44 sequences from tomato, 43 sequences from potato, 37 sequences from capsicum and 28 sequences from coffee (Fig 1B). Unlike the MEKK tree, the Raf tree was divided into 11 different clades, with an uniform clustering of genes in all the clades. The NJ tree for ZIK was generated using 11 sequences from tea, 11 sequences from Arabidopsis, 10 sequences from rice, 10 sequences from tomato, 16 sequences from potato, 6 sequences from capsicum and 8 sequences from coffee (Fig 1C). The ZIK tree was divided into 7 clades and had a uniform clustering of genes in all the clades with only clade E consisting of 2 genes each of Arabidopsis and rice. The results are suggestive of the fact that the genes are either homologous or orthologous

**Table 3. Sequence characteristics and physicochemical properties of MAPKKKs belonging to ZIK subfamily in *C. sinensis*.**

| Gene ID | Locus position | Gene length (bp) | Protein length (aa) | Mol. Wt. (kDa) | pI value | No. of negative residues | No. of positive residues | GRAVY index | Instability index | Aliphatic index | Subcellular localization |
|---------|----------------|------------------|---------------------|-----------------|----------|--------------------------|--------------------------|-------------|-------------------|------------------|--------------------------|
| TEA010125.1 | Scaffold52:664232-675917- | 11685 | 675 | 76706.84 | 5.67 | 91 | 68 | -0.546 | 47.51 | 72.79 | Nucleus |
| TEA022762.1 | Scaffold9600:223976–236388+ | 12412 | 732 | 83655.02 | 5.87 | 103 | 83 | -0.419 | 50.69 | 89.07 | Nucleus |
| TEA024720.1 | Scaffold1050:31289–41391+ | 10102 | 655 | 74571.17 | 6.33 | 89 | 81 | -0.518 | 51.00 | 77.68 | Nucleus |
| TEA002087.1 | Scaffold754:205321-211058- | 5737 | 719 | 81783.51 | 5.54 | 108 | 80 | -0.582 | 42.80 | 75.41 | Nucleus |
| TEA013346.1 | Scaffold5883:262251–272009+ | 9758 | 831 | 94422.51 | 5.53 | 124 | 93 | -0.504 | 43.60 | 79.06 | Nucleus |
| TEA013344.1 | Scaffold5883:191507–194485+ | 2978 | 481 | 55933.38 | 5.65 | 72 | 58 | -0.472 | 40.70 | 80.04 | Nucleus |
| TEA031068.1 | Scaffold1571:837990–857219+ | 19229 | 762 | 86343.00 | 5.92 | 98 | 76 | -0.375 | 40.49 | 85.07 | Chloroplast |
| TEA020698.1 | Scaffold2762:535605-540535- | 4930 | 664 | 75716.56 | 5.63 | 95 | 74 | 0.439 | 46.00 | 81.91 | Nucleus |
| TEA027328.1 | Scaffold688:688353–693133+ | 4780 | 748 | 84782.46 | 5.48 | 100 | 81 | -0.350 | 42.68 | 80.16 | Chloroplast |
| TEA020112.1 | Scaffold1093:624579–626760+ | 2181 | 300 | 34181.96 | 5.60 | 47 | 40 | -0.428 | 33.33 | 85.47 | Nucleus |
| TEA033250.1 | Scaffold4160:2129637–2136050 + | 6415 | 622 | 69665.68 | 5.14 | 99 | 74 | -0.449 | 43.80 | 84.00 | Nucleus |

Locus position, gene length, protein length, molecular weight and pI value, no. of negative and positive residues, GRAVY index, instability index, aliphatic index and subcellular localizations were analysed.

to each other. However, the Raf and ZIK genes did not feature any orthologous gene with respect to Arabidopsis. This was validated only when the Raf and ZIK gene accession IDs were scanned in the TPIA database to search for the presence of orthologous genes in the later part of the study (Functional Interaction Network).

## Domain analysis of tea MAPKKKs

Among the 3 subgroups of plant MAPKKKs, the MEKK subfamily is fairly well known and characterized. Most MEKKs are known to be a part of the recognized MAP Kinase cascades, which activates the downstream MKKs. MEKK1 and MEKK2 from Arabidopsis, have been proven to play a significant role in plant innate immunity [33–35]. Similar to other plant MAPKKKs, 16 out of 17 members of MEKK subfamily in tea displayed a characteristic conserved signature G(T/S)Px(W/Y/F)MAPEV, except TEA014429.1 (Fig 2A). Two of the most widely studied Arabidopsis Raf subfamily MAPKKKs, namely CTR1 and EDR1 are known to actively participate in ethylene mediated signalling and defense response mechanisms [8]. All 31 members of the Raf subfamily in tea featured a conserved GTxx(W/Y) MAPE signature in its kinase domain with no exceptions (Fig 2B). The ZIK-like MAPKKKs are also known by the name WNK or with no lysine (K). They are not proven to be involved with the phosphorylation of the MKKs in plants but have specific functions. Arabidopsis ZIK1 is known to phosphorylate APRR3 *in-vitro*, which is a putative component of the circadian clock in plants and is believed to be involved in signal transduction pathway, regulating its biological activity [36]. Another ZIK cascade, involving ZIK2, ZIK5 and ZIK8 in Arabidopsis is known to regulate the flowering time by modulating the photoperiod [37]. The ZIK subfamily featured a characteristic GTPEFMAPE(L/V/M)(Y/F/L) conserved signature across all its 11 members in tea (Fig 2C) [5, 6]. The presence of these distinctive conserved signatures across the tea MAPKKKs further confirms identity and the subfamily they belong. The largest subfamily was found to be the Raf subfamily with 31 members, while the smallest was found to be the ZIK subfamily with only 11 members. These results are consistent with published reports on other plant MAPKKKs [5, 6].

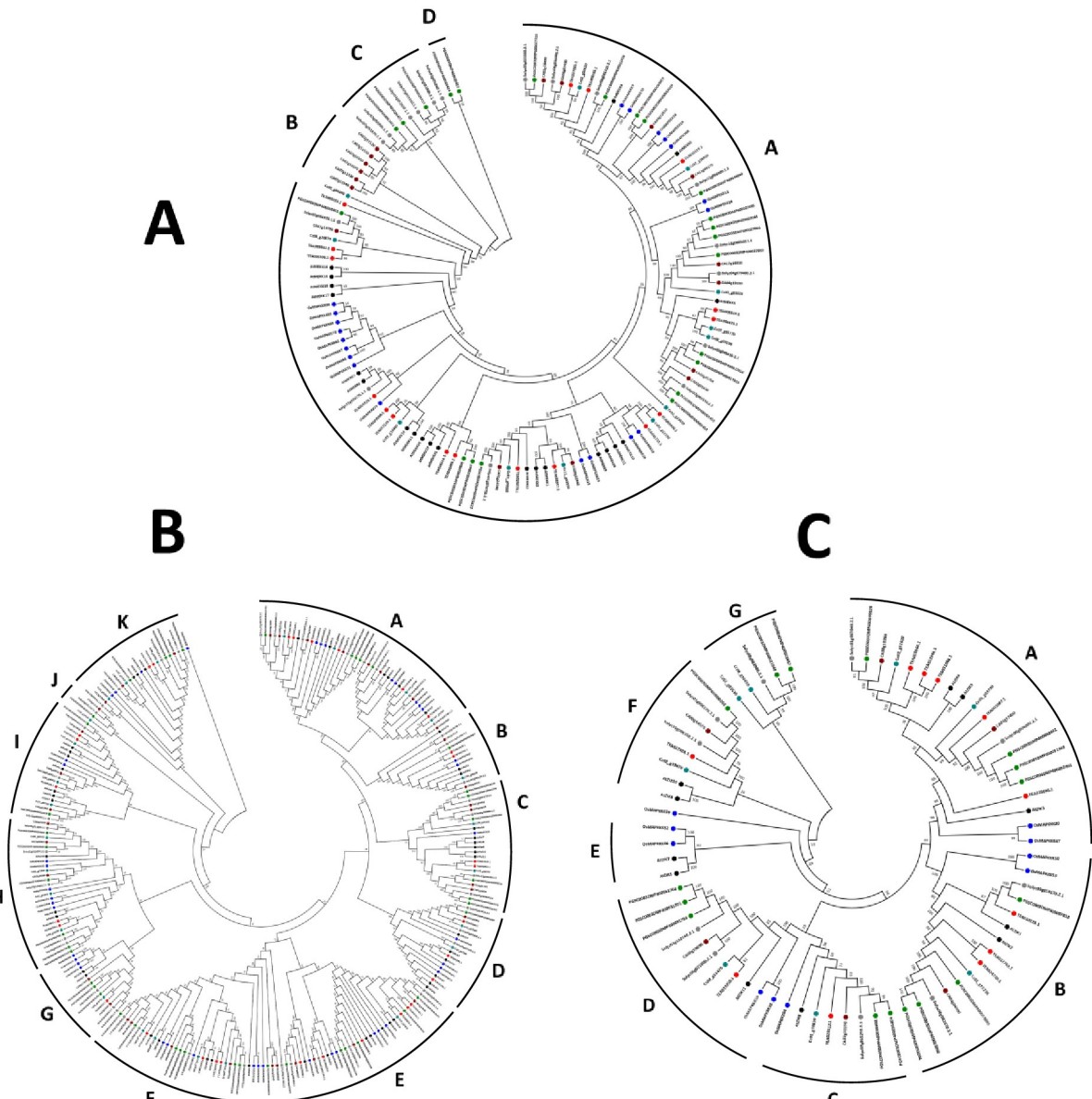

**Fig 1.** Phylogenetic tree of (A) MEKK-like (B) Raf-like and (C) ZIK-like genes from *Arabidopsis thaliana* (black), *C. sinensis* (red), *Oryza sativa* (blue), *Solanum lycopersicum* (grey), *Solanum tuberosum* (green), *Capsicum annum* (brown), *Coffee canephora* (teal). The full-length MEKK, Raf and ZIK protein sequences were aligned using Clustal W, and the phylogenetic trees were constructed using MEGA 7.0.14 by the Neighbor-Joining (NJ) method with default parameters and 1000 bootstrap replicates.

## Motif composition of tea MAPKKKs

To understand the evolution and comprehend sequential characteristics of the MAPKKK proteins in tea, a conserved motif search was carried out using the MEME suite (Fig 3). Ten conserved motifs were identified in each of the three subfamilies. Almost all the tea MAPKKK proteins featured the protein kinase domain with motif 1, motif 2 and motif 3. Motif 4 was conserved across all the proteins with only one exception of TEA031230.1. Motif 5, motif 7 and motif 8 were only obtained for the ZIK subfamily with one exception of a MEKK-like TEA014429.1, which featured motif 8. Motif 6 and motif 9 were harboured by almost all the

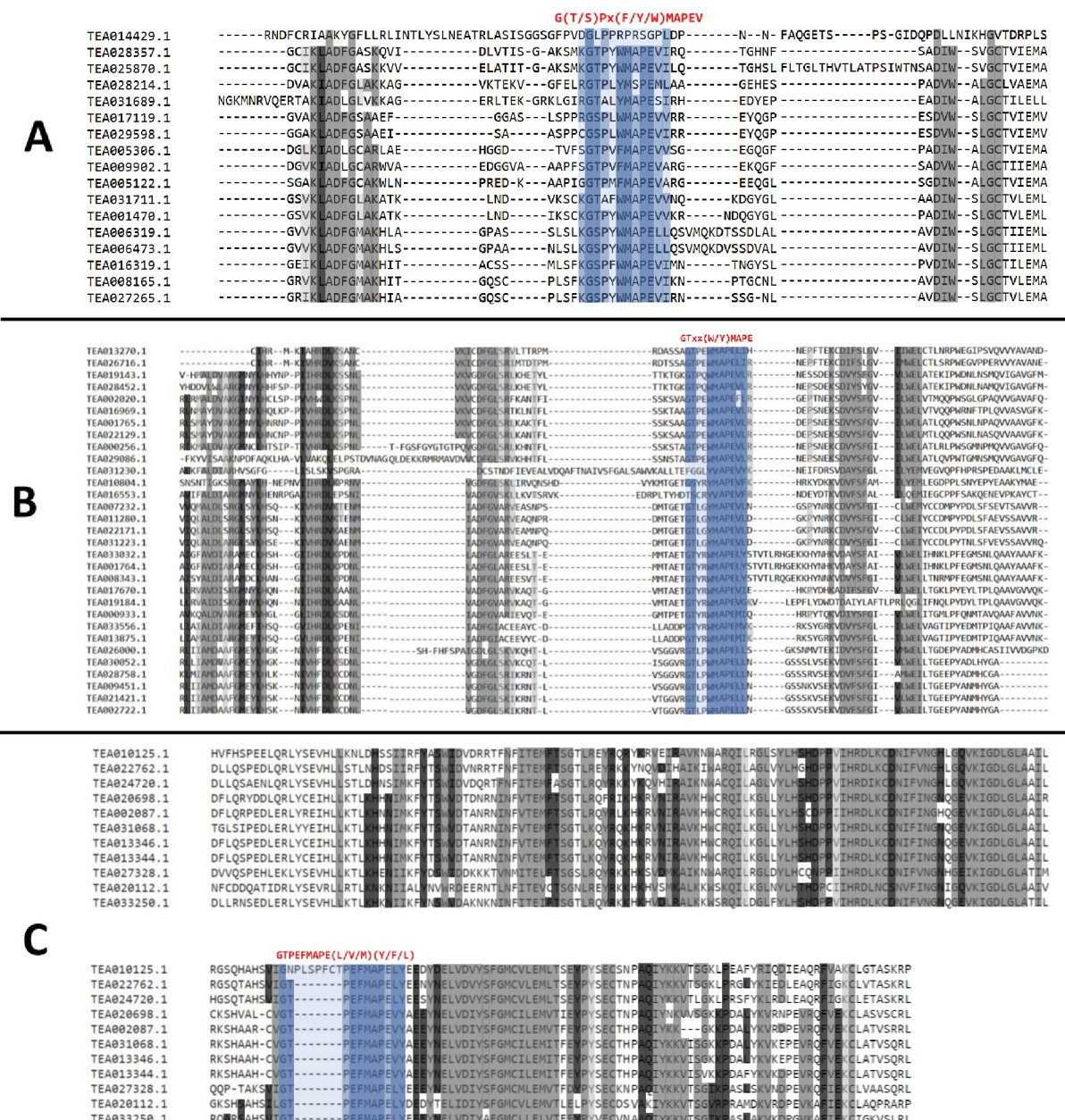

**Fig 2.** Alignment of MAPKKKs of (A) MEKK subfamily (B) Raf subfamily and (C) ZIK subfamily in *C. sinensis*. ClustalX program was used for aligning the obtained sequences. The highlighted part (G(T/S)Px(F/Y/W)MAPEV) shows the conserved signature for the MEKK proteins. The highlighted section (GTxx(W/Y)MAPE) shows the conserved signature for the Raf proteins and the highlighted part (GTxx(W/Y)MAPE) shows the conserved signature for the ZIK proteins.

protein sequences. However, motif 10 was only specific to the MEKK and Raf subfamilies. Motif annotation revealed that motif 2 harboured a protein kinase ATP-binding site. Motif 6 contained a tyrosine kinase phosphorylation site. Motif 9 featured a serine/threonine protein kinase activation site (S4 Fig in S2 File). The results suggested that proteins belonging to a same group harboured similar conserved motifs, further indicating that the classification of the tea MAPKKK subfamilies was backed by motif analyses.

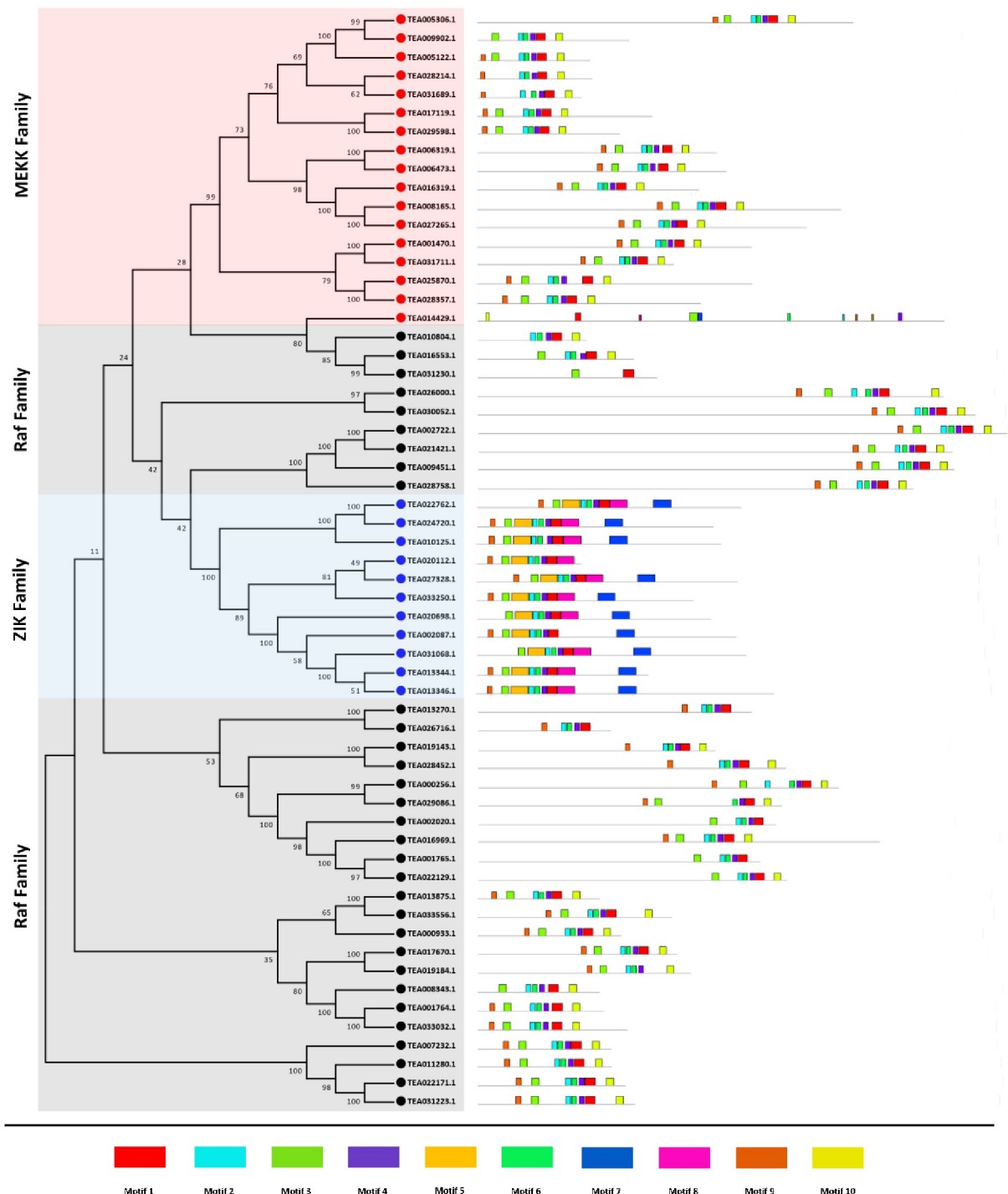

**Fig 3. The motif analysis of 59 identified MAPKKKs in *C. sinensis*.** The motif figures were generated by MEME suite. A total of 10 motifs were identified and are marked individually.

## Gene structure analysis of tea MAPKKKs

The intron-exon distribution pattern for tea MAPKKKs were analysed and visualised using the Gene Structure Display Server v2.0. Study of gene structure revealed differences in number

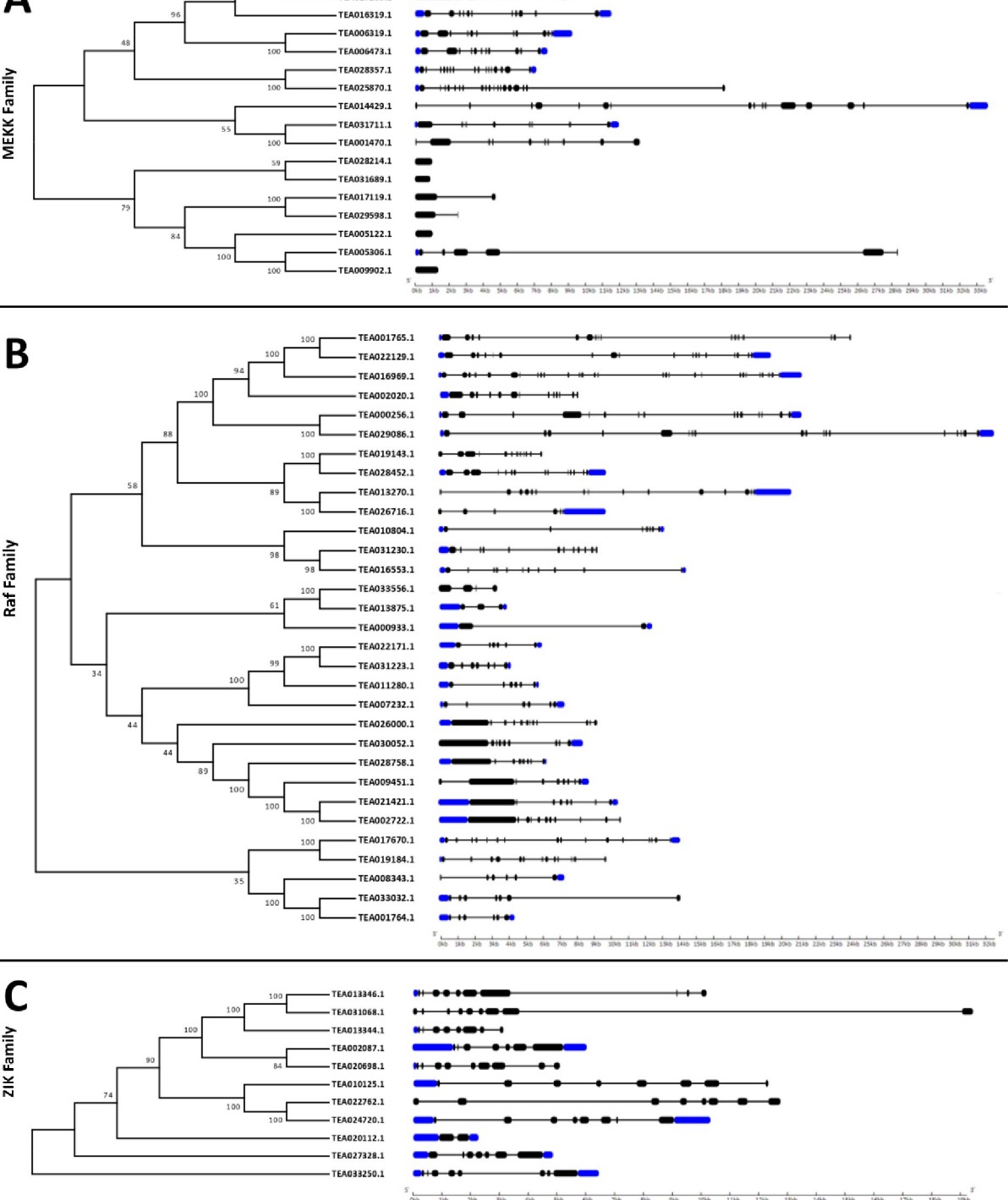

**Fig 4.** The intron/exon architecture of (A) MEKK (B) Raf and (C) ZIK genes in *C. sinensis*. Gene structure maps were drawn using the Gene Structure Display Server 2.0. Black boxes represent exons, blue boxes represent the UTRs and black lines represent introns. The gene length can be estimated by using the scale (in kb) given at the bottom.

of introns and exons, which contributes to variation in gene length. Introns or non-coding sequences are found abundantly within a genome and are regarded as an indicator of genome complexity [38, 39]. Analysis of the intron patterns could help to comprehend and provide insights into the evolution, function and regulation of the genes [38, 40–43]. The analysis of the intron-exon architecture in tea revealed significant variation in the number of introns and exons among the three subfamilies of MAPKKKs (Fig 4). However, genes belonging to the same clades had similar intron-exon distribution. The MEKK subfamily had 10 out of 17 genes (59% of the MEKK genes) possessing 6 to 16 exons (Fig 4A). TEA025870.1 had 19 exons and 18 introns in its gene. Two genes possessed 2 exons and 1 intron and the remaining 4 genes had no introns. Only 9 out of 17 genes featured UTR (Untranslated Regions) segments and 5 out of these 9 genes featured both 5' and 3' UTRs. 3 genes contained only the 5' UTR segments and 1 gene only had the 3' UTR segment. The genes belonging to the Raf subfamily had exons ranging from 6 to 18 and was featured by 27 out of 31 genes (87% of the Raf genes) (Fig 4B). TEA016969.1 featured a staggering 28 exons and was the highest among all the Raf genes. Three genes namely TEA000933.1, TEA013875.1 and TEA033556.1 had 2, 3 and 4 exons respectively which were the lowest number of exons found amongst all the Raf genes. 29 out of 31 genes possessed UTR segments. However, only 17 of the 29 genes had both 5' and 3' UTRs. 7 genes featured only the 5' UTR segment and remaining 5 genes only had the 3' UTR. Unlike the MEKK and Raf subfamilies, ZIK subfamily displayed a certain level of conservancy with respect to the number of exons and introns. 10 out of 11 ZIK genes (91% of the ZIK genes) had exons ranging from 7 to 10 (Fig 4C). TEA020112.1 however featured only 2 exons. 9 out of 11 genes possessed UTR segments and 5 of them had both 5' and 3' UTRs. 4 genes featured only the 5' UTR segment. However, no ZIK subfamily gene in tea featured only the 3' UTR segment like the MEKK and Raf subfamilies.

## Retrieval of promoter sequences and analysis of cis-regulatory elements

Cis-acting regulatory elements are often used for determining the function of genes, regulation of gene transcription and gene expression [44, 45]. In order to explore the transcriptional regulation and putative functions of the tea MAPKKK genes, promoter sequences of 2000 bp upstream of the initiation codon "ATG" was retrieved from the TPIA database. These sequences were then analysed using the PlantCARE database for the identification of the cis-acting regulatory elements (CAREs). It was found that the cis-acting elements were randomly scattered in the promoter regions of the tea MAPKKKs. The study revealed an aggregate of 56 CAREs in all the tea MEKK, Raf and ZIK genes (S4 Table in S1 File). These elements were also arranged and grouped based on their specific biological functions (Fig 5A). The sequence length of the cis-acting elements ranged from 5 to 14 bp (Fig 5B) with most of the CAREs having sequence lengths of 6 and 9 bp. The analysis of the 56 CAREs revealed the involvement of 13 elements in plant growth and regulation, 26 in light responsiveness, 6 were stress response elements and the remaining 11 were involved in phytohormone responses. The light responsive elements comprised of the largest section of the identified CAREs in all of the 59 tea MAPKKKs with 26 regulatory elements. Among these, Box-4 and G-Box accounted for the major part in 51 and 45 tea MAPKKKs. Some of the other light response elements included AE box, GATA motif, GT1-motif, MRE and TCT motif. The CAREs related to the phytohormone responses mainly involved abscisic acid responsive element (ABRE), and MeJA responsive elements (CGTCA-motif and TGACG-motif) in 42 and 38 tea MAPKKKs accordingly. Other phytohormone responsive elements comprised of gibberellin responsive elements (TATC-box, P-box and GARE-motif), auxin responsive elements (TGA-element, AuxRR-core, TGA-box and AuxRE) and salicylic acid responsive element (TCA-element) in 35, 33

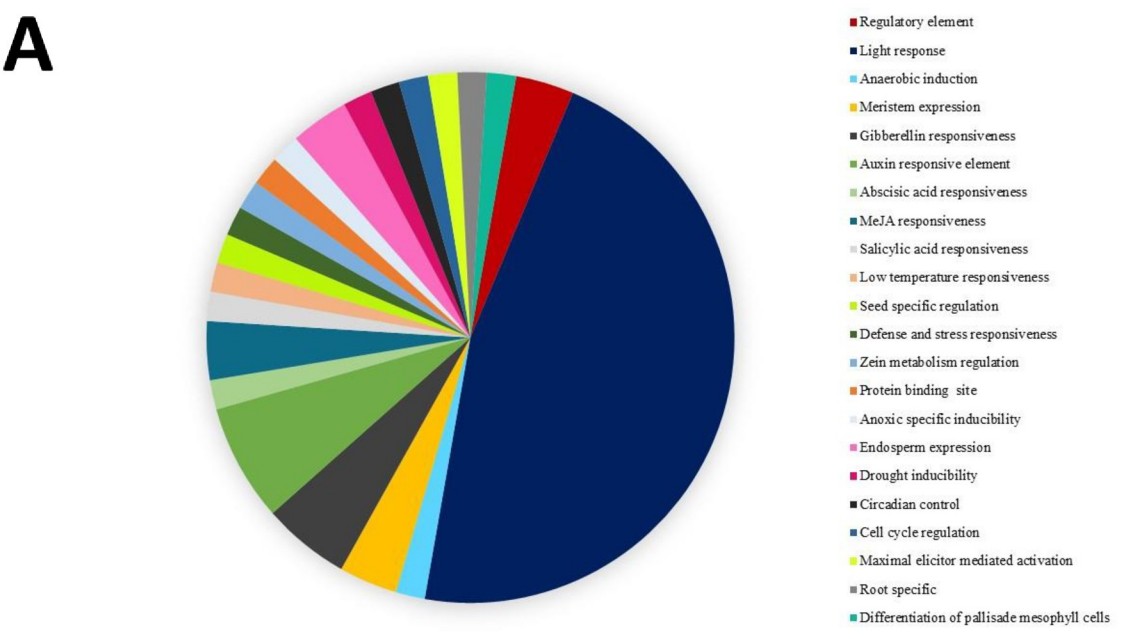

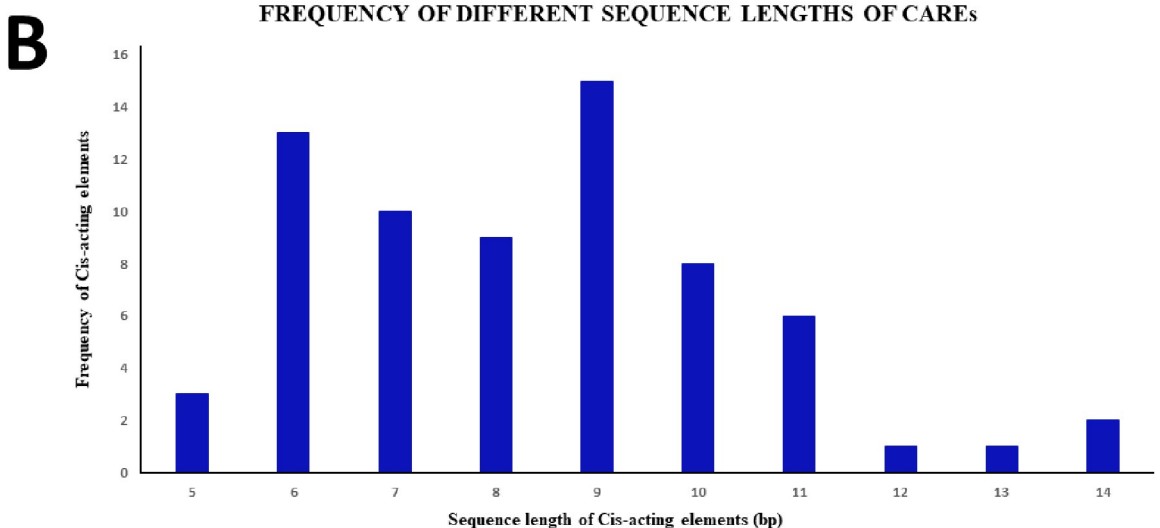

**Fig 5. Analysis of cis-acting elements identified from the MEKK, Raf and ZIK genes of *C. sinensis*.** All cis-acting elements have been identified using PlantCARE database. (A) Pie-chart showing the frequency of different cis-acting elements based on their specific biological activities. (B) Histogram showing the frequency of different sequence lengths of the cis-acting elements.

and 26 tea MAPKKKs respectively. Among the cis-elements which are associated with plant growth and development, 21 tea MAPKKKs possessed meristem expression elements (CAT-box and NON box) while 18 genes had zein metabolism regulatory element (O2-site). Other plant growth related CAREs included regulatory elements (A-box and Box-II like sequences), endosperm expression element (GCN4_motif), circadian control, cell cycle regulation (MSA-like), Box-III and few seed specific (RY-element), root specific (motif I) and palisade

mesophyll differentiation (HD-Zip 1) element that were discovered on the promoter regions of 19, 12, 8, 3, 7, 2, 1 and 1 tea MAPKKKs respectively. In addition, numerous stress response CAREs were also identified in the promoter regions. These included ARE (anaerobic induction element), LTR (low temperature responsiveness), TC-rich repeats (defense and stress responsiveness), MBS (drought inducibility), GC-motif (anoxic specific inducibility) and AT-rich sequences (maximal elicitor mediated activation) in 52, 19, 25, 18, 8 and 6 tea MAPKKKs respectively. These results indicate the involvement of MAPKKK genes in various responses like phytohormone treatments, low temperature, physiological stresses and plant growth and regulation.

## Genomic distribution map and evolutionary pressure on tea MAPKKKs

The tea MAPKKKs were mapped onto the genomic scaffolds to understand their distribution pattern. Due to the lack of chromosome-level assembly data in the TPIA database, the genes were mapped onto their respective scaffolds instead of the chromosomes. All 59 tea MAPKKKs were extensively distributed across 58 different genomic scaffolds. 17 MEKK genes were distributed across 17 different scaffolds (Fig 6A). Similarly, 31 Raf genes were distributed across 31 genomic scaffolds (Fig 6B). 11 ZIK genes were mapped onto 10 genomic scaffolds (Fig 6C). Two ZIK genes namely, TEA013344.1 and TEA013346.1 were mapped on the same genomic scaffold 5883 and thus featured a duplication event. Additionally, both these genes possessed similar intron-exon architecture. This result is conclusive evidence that duplication events were of significant importance and played a crucial role in the expansion of the MAPKKK genes in *C. sinensis* genome. Further, the ratio of non-synonymous substitution rates ($K_a$) and synonymous substitution rates ($K_s$) was evaluated to illuminate the mechanism of gene divergence and evolutionary pressure on the tea MAPKKKs. The ratio determines the selective pressure acting on the respective proteins. If the $K_a/K_s$ ratio is <1, it determines negative or purifying selection. If the $K_a/K_s$ ratio is = 1, it indicates neutral selection and if the $K_a/K_s$ ratio is >1, it signifies positive selection [46]. For the MEKK subfamily, pair wise comparisons revealed that 72 gene pairs had $K_a/K_s$ ratios above 1, indicating that they are under positive selection, 24 gene pairs had values less than 1, indicating a negative selection and remaining 40 were not a number (Nan) (S5 Table in S1 File). Similarly, $K_a/K_s$ ratios of the Raf subfamily revealed 341 gene pairs in positive selection, 96 in negative selection and 28 pairs as Nan (S6 Table in S1 File). $K_a/K_s$ ratios of ZIK subfamily uncovered 30 pairs in positive selection, 21 in negative selection and the remaining 4 as Nan (S7 Table in S1 File). The $K_a/K_s$ cumulative graphs of tea MAPKKKs were also generated (S5-S7 Figs in S2 File). The results suggest strong positive selection pressures would have occurred, enabling different factors to regulate the MAPKKKs in *C. sinensis*.

## GO ontology analysis and functional interaction network of tea MAPKKKs

The GO ontology analysis was performed in order to predict the potential functions of all the 59 tea MAPKKKs (S8 Fig in S2 File). All the MEKK, Raf and ZIK proteins were assigned into three major ontologies and 14 GO terms. The statistical significance of these GO terms were expressed in terms of p values (S8 Table in S1 File). The 3 major ontologies were biological process, cellular component and molecular function. In biological process, the proteins were distributed into 6 GO terms with 'protein phosphorylation' (GO:0006468, 23 sequences, 38.98%) with the largest representation. In the cellular component group, the MAPKKK proteins were distributed into 3 GO terms. Among these 3, 'cytosol' (GO:0005829, 5 sequences, 8.47%) had the highest representation followed by 'cytoplasm' (GO:0005737, 3 sequences, 5.08%) and 'intracellullar' (GO:0005622, 3 sequences, 5.08%). The molecular function

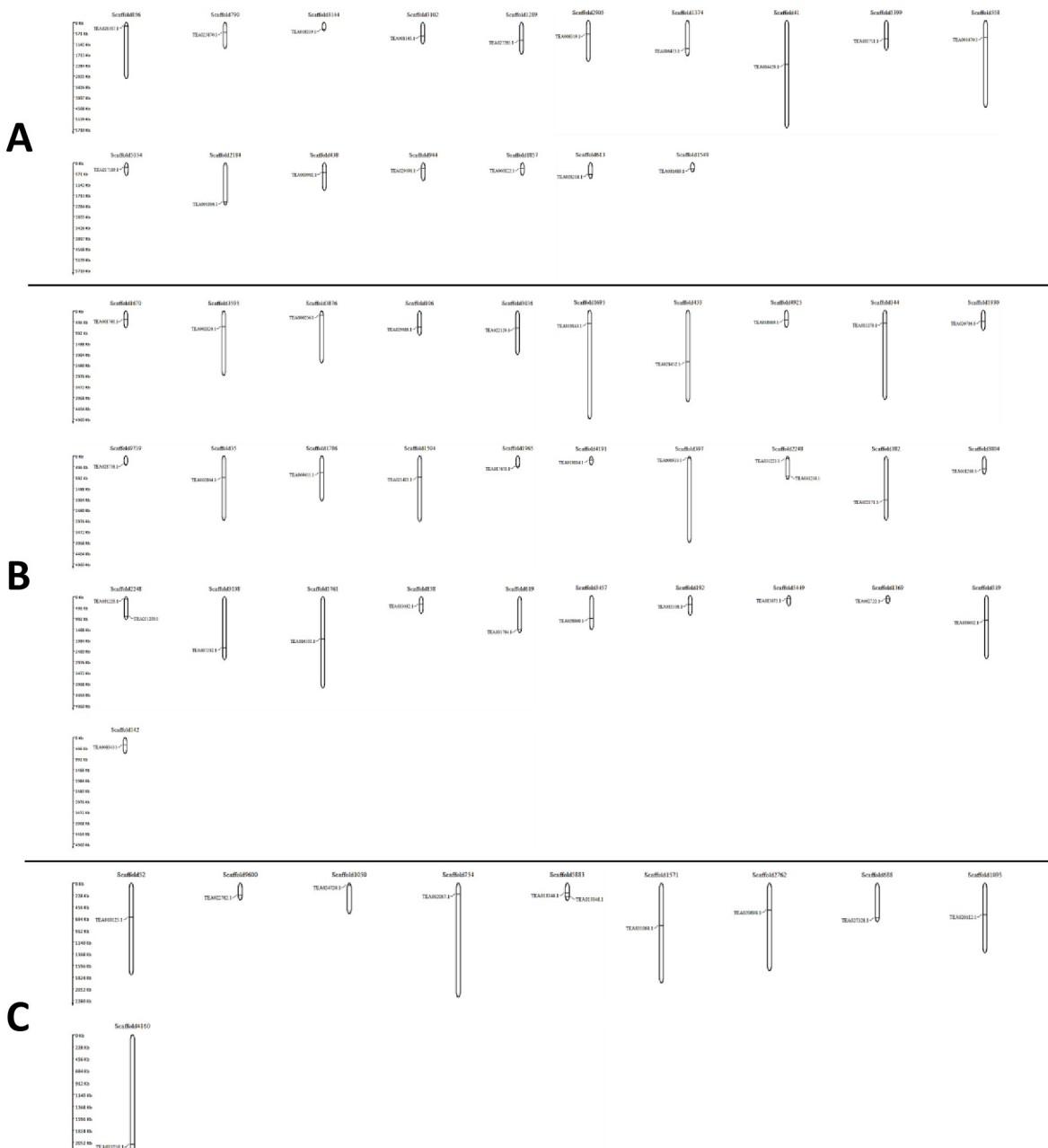

**Fig 6.** The scaffold distribution of (A) MEKK subfamily (B) Raf subfamily and (C) ZIK subfamily genes in *C. sinensis*. MapGene2chromosome web v2 (MG2C) software tool (http://mg2c.iask.in/mg2c_v2.1/) was used to map genes onto their respective scaffolds. The scaffolds are drawn to scale and the scaffold numbers are indicated on the top.

ontology featured 5 GO terms with 'protein serine/threonine kinase activity' (GO:0004674, 16 sequences, 27.11%) and 'kinase activity' (GO:0016301, 11 sequences, 18.64%) having highest representation. They were followed by 'receptor signaling protein serine/threonine kinase activity' (GO:0004702, 5 sequences, 8.47%), 'MAP kinase kinase kinase activity' (GO:0004709, 4 sequences, 6.78%) and 'protein tyrosine kinase activity' (GO:0004713, 3 sequences, 5.08%).

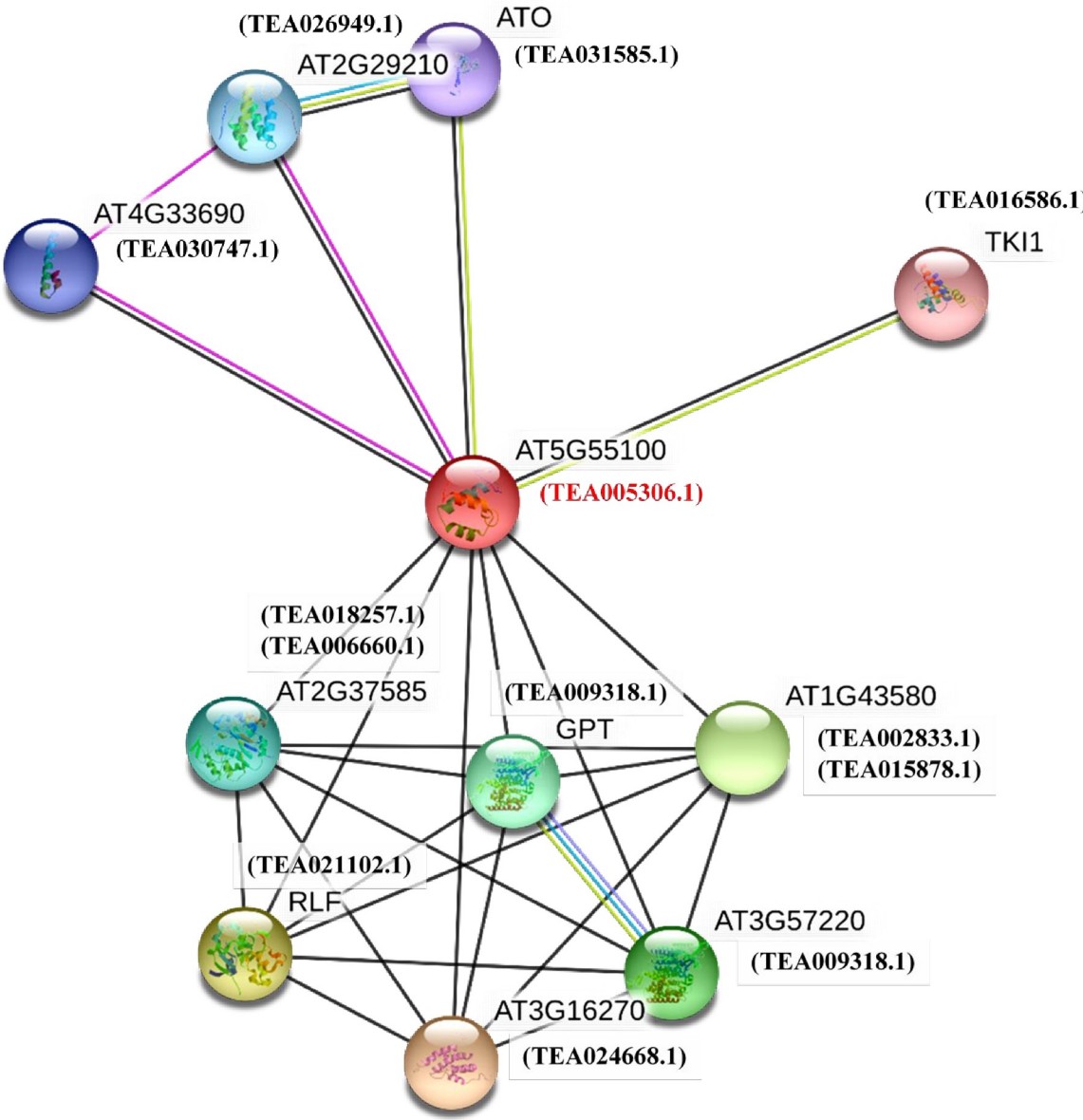

**Fig 7. Functional interaction network of tea MAPKKK proteins.** The interaction network was build according to the ortholog in Arabidopsis. TEA005306.1 in tea is orthologous to AT5G55100 in Arabidopsis. The orthologous protein (red) and homologous proteins (black) are shown within brackets.

For better understanding of the interactions of MAPKKKs in *C. sinensis*, an interaction network was constructed based on the orthologous genes in Arabidopsis, using the STRING server (Fig 7). The functional interaction network of the genes has been built using that of Arabidopsis since tea database is not included in the STRING online server. All the identified tea MAPKKKs were searched in the TPIA database to find the presence of any orthologous gene and TEA005306.1 was found to be orthologous to AT5G55100 in Arabidopsis. AT5G55100 was then used to build the interaction network. Since, similarity search programs like BLAST are widely used to produce accurate statistical estimates [47] and proteins that have high sequence and structural similarity generally tend to possess similar functions [48] therefore,

tea proteins homologous to the Arabidopsis proteins were incorporated in the network (Fig 7). This was done to predict the functional interaction network of TEA005306.1 in the tea genome, *in-silico* based on the Arabidopsis association model. These homologous proteins were designated as STRING proteins and were selected on the basis of high bit scores. Based on TAIR database, AT5G55100 is involved in RNA processing and is expressed during 15 growth stages in 24 different organs and tissue of plant. It shows interactions with AT4G33690 which is involved in biological process of protein binding [16]. AT2G29210 is involved with RNA splicing, mRNA processing and is expressed during 13 different growth stages in 23 different organs and tissues [16]. ATO (AT5G06160) encodes for a protein similar to pre-mRNA splicing factor SF3a60 and is involved in gametic cell fate determination [16]. Loss of function results in the ectopic expression of egg cell makers, thereby suggesting a role in restriction of gametic cell fate. TEA031585.1, which is homologous to ATO gene, is a part of the spliceosomal complex and is involved in mRNA splicing based on GO ontology. TK1 (AT2G36960) is a TSL-kinase interacting protein and is involved in protein binding [16]. It is expressed in 14 developmental stages in 25 different plant organs and tissue. GPT (Glucose-6-Phosphate translocator) (AT2G41490) is an integral component of membrane and has a UDP-N-acetylglucosamine-dolichyl-phosphate N-acetylglucosamine phosphotransferase activity [16]. It is expressed during 15 developmental stages in 23 different organ and tissue in the plant. AT3G57220 is located in the endoplasmic reticulum and has a UDP-N-acetylglucosamine-dolichyl-phosphate N-acetylglucosamine phosphotransferase activity. It is also linked with polysaccharide biosynthesis and is expressed during 10 growth stages in 16 different plant tissue and organ [16]. According to GO ontology, TEA009318.1 is also involved in phosphotransferase activity in tea and is homologous to both AT2G41490 and AT3G57220 in Arabidopsis.

## Tissue specific gene expression of tea MAPKKKs

The tissue specific expression pattern of the tea MAPKKK genes in various plant tissues were retrieved from the TPIA database where levels of expression were expressed using transcripts per million (TPM). The TPIA database houses tissue specific expression data for 8 different plant tissues which includes apical bud, flower, fruit, young leaf, mature leaf, old leaf, root and stem (S9 Table in S1 File). Among the 59 tea MAPKKK genes, expression data for 58 genes were retrieved with an exception of 1 MEKK gene, TEA031689.1. All 58 genes displayed varied levels of expression, with few of the transcripts barely readable (Fig 8). For the MEKK genes, the maximum level of expression in apical bud was shown by TEA006319.1. This gene also marked the highest level of expression in young leaf. TEA017119.1 showed highest level of expression in flower. TEA016319.1 displayed highest expression levels in fruit, mature leaf, old leaf and stem. TEA005122.1 was expressed maximum in root. TEA028357.1 and TEA009902.1 had negligible levels of expression in all of the 8 plant tissues (Fig 8A). For the Raf genes, TEA000933.1 showed highest levels of expression in apical bud, fruit, young leaf, mature leaf, old leaf, root and stem. TEA007232.1 was expressed maximum in flower. However, TEA001765.1, TEA013270.1, TEA028758.1 and TEA031230.1 had negligible levels of expression (Fig 8B). Finally, for the ZIK genes, TEA002087.1 displayed highest levels of expression in apical bud, flower, young leaf and stem. TEA022762.1 had highest levels of expression in fruit, mature leaf and old leaf. TEA020112.1 showed maximum expression in root. However, TEA013344.1, TEA031068.1, TEA020698.1 and TEA027328.1 showed minor levels of expression (Fig 8C). Heat maps for all the 58 genes, representing the tissue specific expression levels were also being generated (S9 Fig in S2 File).

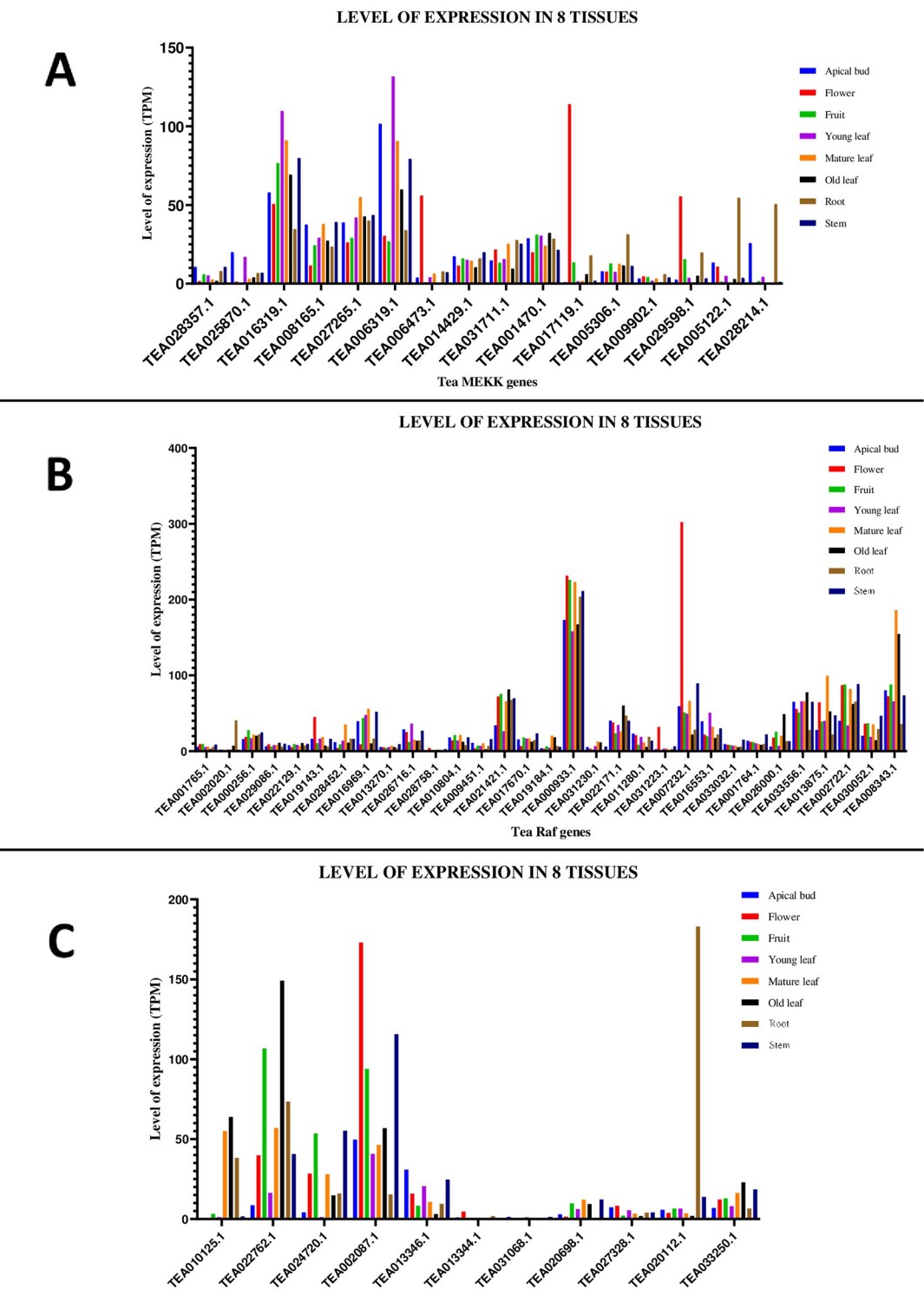

**Fig 8.** Tissue-specific expression patterns of (A) MEKK (B) Raf and (C) ZIK genes in *C. sinensis*. The relative expression of these genes were analysed in different tissues by using GraphPad Prism 8 software. The level of expression was in transcript per million (TPM). 58 out of the 59 identified genes had expression data in TPIA database with an exception of 1 MEKK gene (TEA031689.1).

## Abiotic stress induced differential expression levels of tea MAPKKKs

The expression data of the leaves of the tea plant present in TPIA database (S10-S13 Tables in S1 File) was used in order to study the effect of cold, drought, salt stress along with methyl jasmonate (MeJA) treatment followed by the generation of expression graphs for the same (Figs 9–12). The cold acclimated (CA) data (unpublished), present in the TPIA database consists of 5 stages of expression. These are: 1. 25~20 ˚C (CK), 2. Fully acclimated at 10 ˚C for 6 h (CA1-6h) 3. 10~4 ˚C for 7 days (CA 1-7d), 4. Cold response at 4~0 ˚C for 7 days (CA 2-7d) and 5. Recovering under 25~20 ˚C for 7 days (DA-7d), where CK is the control [49]. Expression of MEKK genes revealed that 15 out of 17 genes were upregulated under CA 1-6h. TEA006473.1 was downregulated while TEA031689.1 displayed no expression levels. Expression levels under the CA 1-7d condition showed that 12 genes were upregulated, 4 genes were downregulated and remaining 1 gene showed no data. Under the CA 2-7d condition, expression levels revealed that 10 genes were upregulated, 6 genes were downregulated and remaining 1 gene displayed no expression data. Lastly, under the DA-7d condition, data revealed that 13 genes showed upregulation, 3 genes showed downregulation and 1 gene had no data (Fig 9A). The Raf and ZIK genes were also analysed based on the same 5 conditions. For the Raf genes, under CA 1-6h condition, 22 genes out of 31 were upregulated and 9 genes were downregulated. Under CA 1-7d condition, 16 genes were upregulated and 15 genes were downregulated. Expression levels under CA 2-7d revealed that 17 genes showed upregulation and remaining 14 genes showed downregulation. Under DA-7d condition, 21 genes were upregulated and 10 genes were downregulated (Fig 9B). Expression data of the ZIK genes revealed that under CA 1-6h, 7 out of 11 genes were upregulated and 4 genes were downregulated. CA 1-7d condition revealed that 5 genes were upregulated, 5 genes were downregulated and remaining 1 gene displayed no expression. Under CA 2-7d condition, 4 genes were upregulated, 6 genes were downregulated and 1 gene had no expression. Finally, under DA-7d, 8 genes showed upregulation and remaining 3 showed downregulation (Fig 9C). Heat maps for the retrieved expression data were also generated (S10 Fig in S2 File).

Further, expression levels of all tea MAPKKKs were checked under drought stress conditions. Expression levels under drought stress are available in the TPIA database with respect to 25% polyethylene glycol (PEG) treatment and it includes 4 different stages: 1. 0h; 2. 24h; 3. 48h; and 4. 72h [50], where 0h was taken as the control. The expression levels of MEKK genes revealed that under PEG-N-24h condition, 12 genes were upregulated, 4 were downregulated and 1 gene did not show any expression. Under PEG-N-48h, 12 genes were upregulated, 4 were downregulated and 1 gene showed no expression. PEG-N-72h revealed 11 genes showing upregulation, 5 genes showing downregulation and 1 gene with no expression (Fig 10A). Expression of Raf genes showed that under the PEG-N-24h condition, 11 genes were upregulated, 20 genes were downregulated. Under PEG-N-48h, 16 genes showed upregulation while the remaining 15 genes were downregulated. PEG-N-72h revealed that 15 genes were upregulated and 16 genes were downregulated (Fig 10B). Finally, the expression data of ZIK genes revealed 10 out of 11 genes had different expression levels under the given conditions while 1 gene (TEA013344.1) had no data. Under the PEG-N-24h condition, expression data showed that only 1 gene was upregulated while the rest of the genes were downregulated. PEG-N-48h condition too revealed the same result with only 1 gene being upregulated. However, PEG-N-72h showed that 2 genes were upregulated and the rest of the genes were downregulated (Fig 10C). Heat maps for the afore-mentioned data were also generated (S11 Fig in S2 File).

The expression levels of the tea MAPKKKs under salt stress condition were studied. Similar to the drought stress parameters, the salt stress data in TPIA database is recorded based on treatment with 200 mM NaCl under 4 stages: 1. 0h; 2. 24h; 3. 48h; and 4. 72h where 0h was

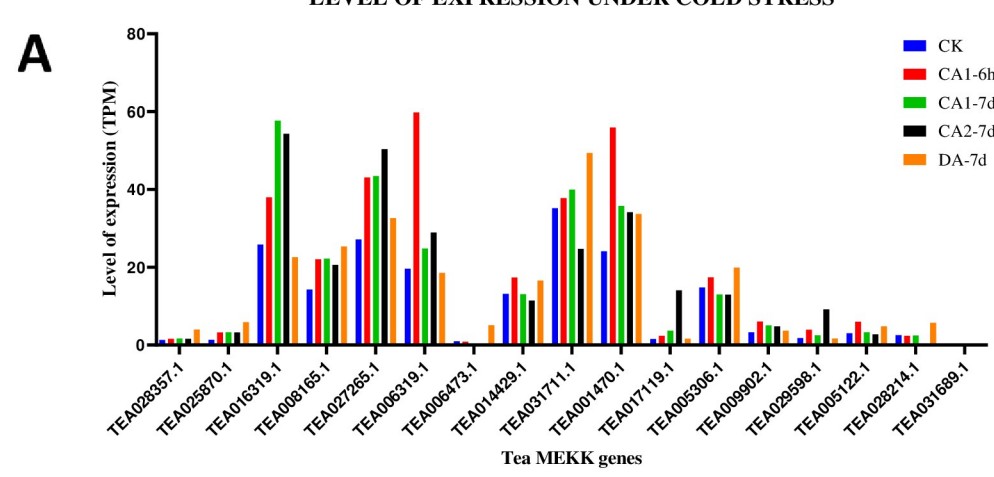

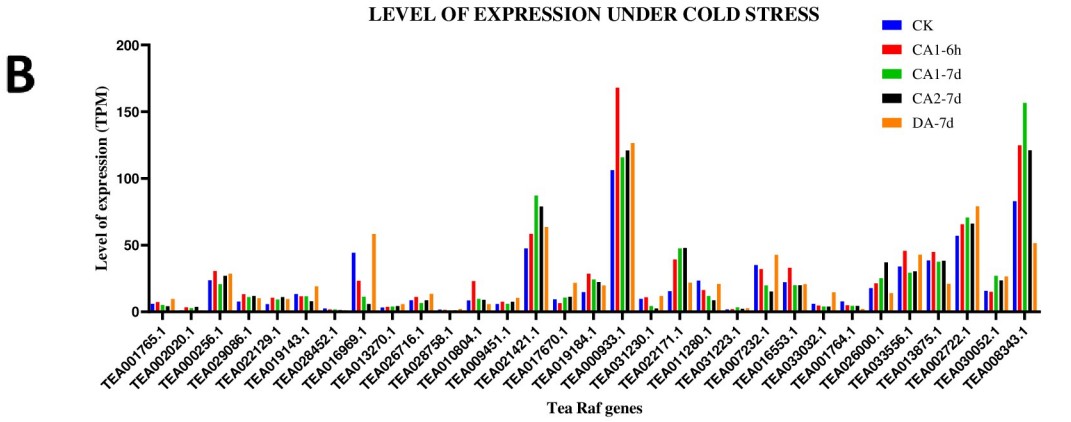

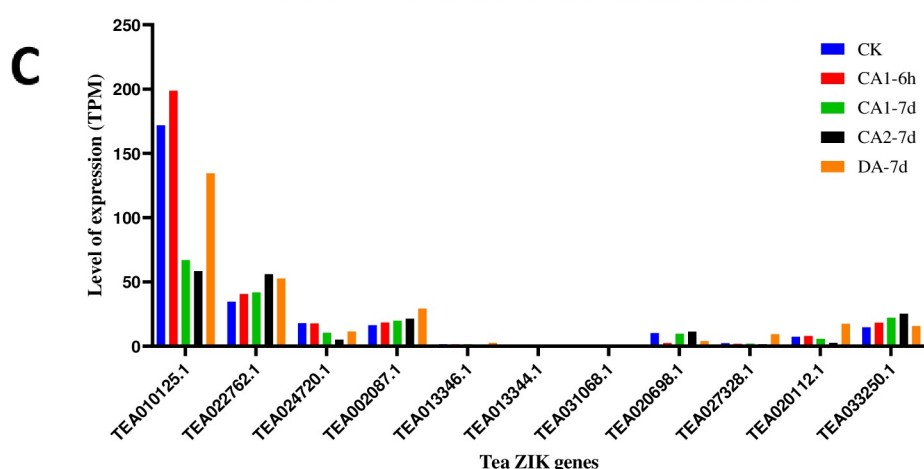

**Fig 9.** Gene expression patterns of (A) MEKK (B) Raf and (C) ZIK genes, under cold stress conditions in *C. sinensis*. The relative expression of these genes were analysed in different stages by using GraphPad Prism 8 software. The level of expression was in transcript per million (TPM).

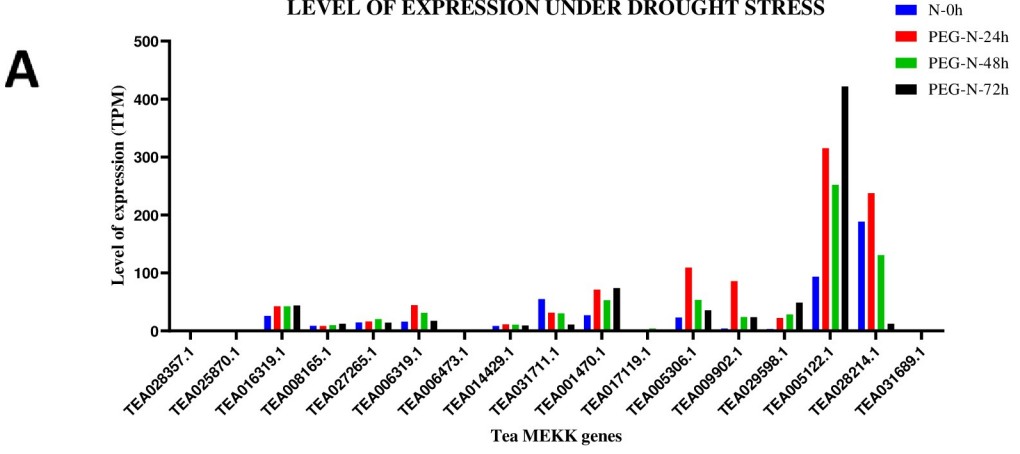

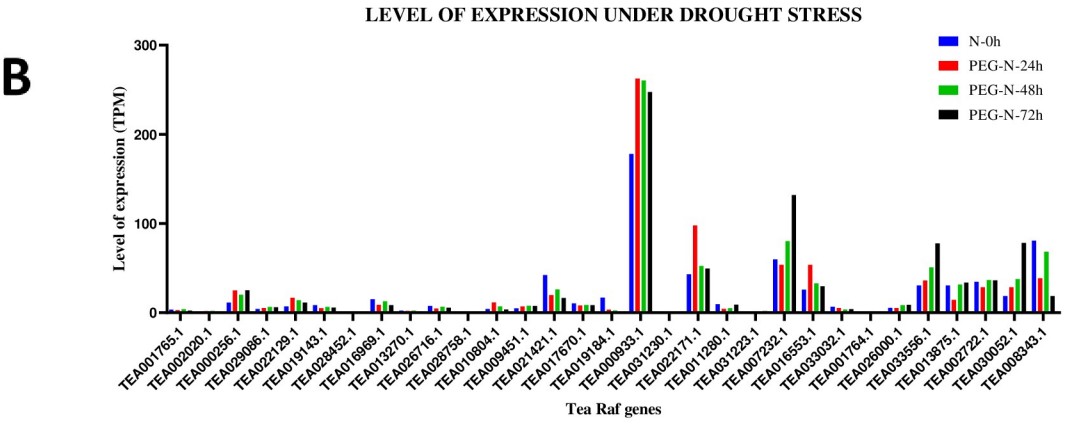

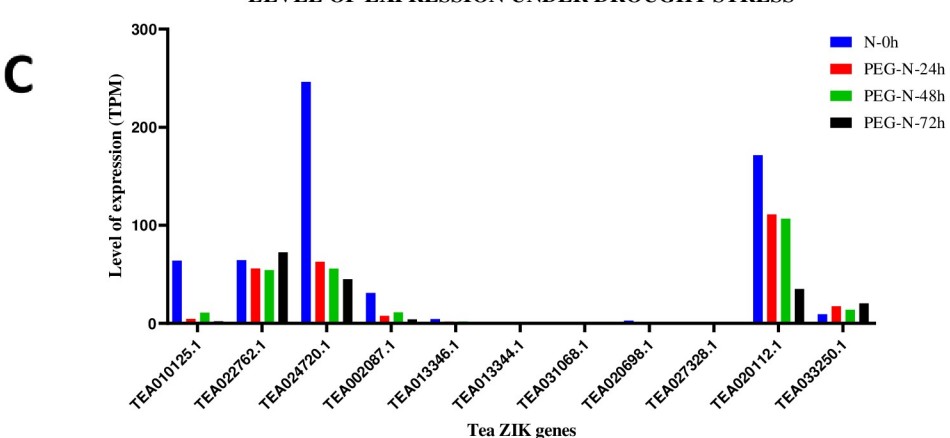

**Fig 10.** Gene expression patterns of (A) MEKK (B) Raf and (C) ZIK genes, under drought stress conditions in *C. sinensis*. The relative expression of these genes were analysed in different stages by using GraphPad Prism 8 software. The level of expression was in transcript per million (TPM).

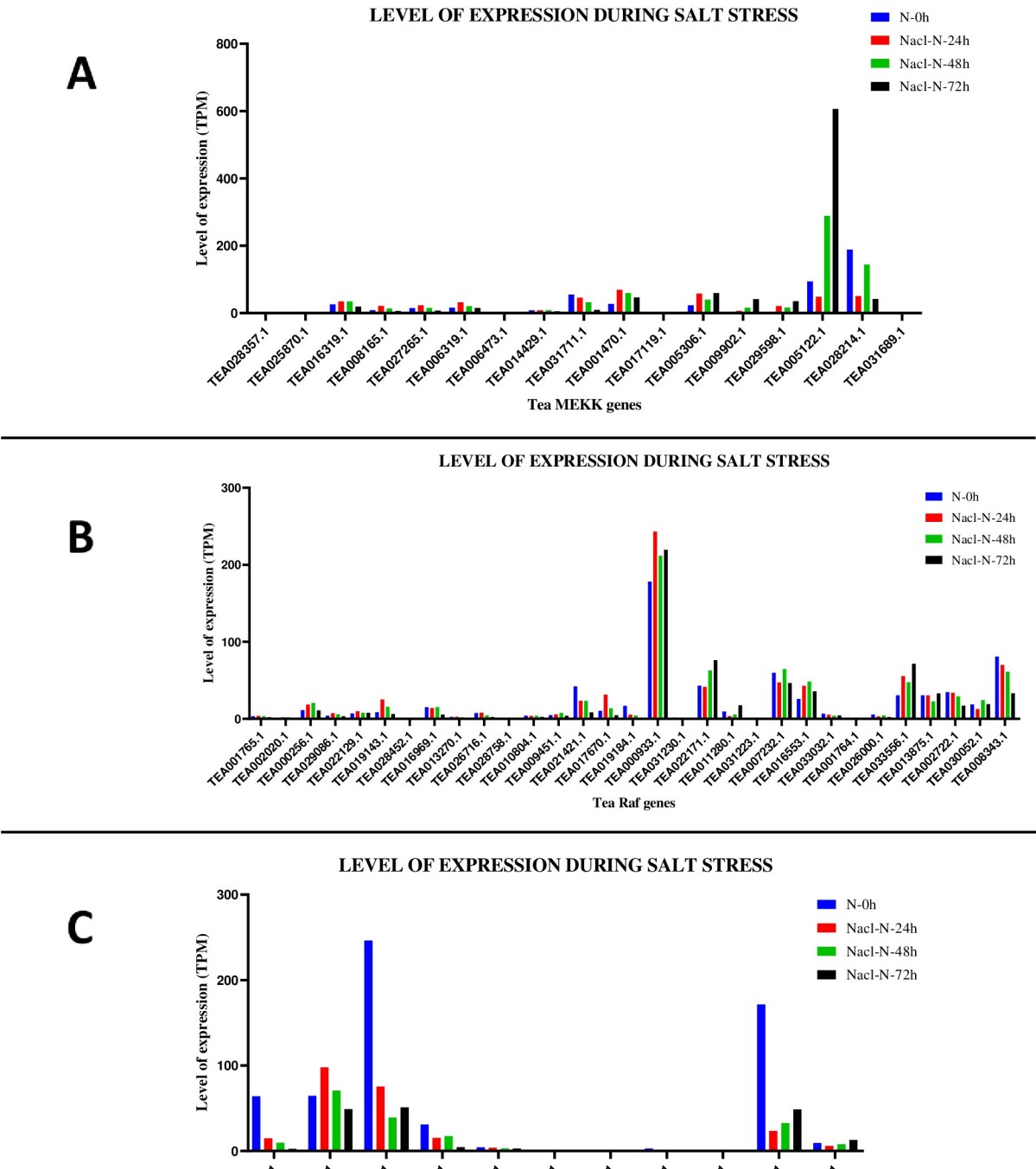

**Fig 11.** Gene expression patterns of (A) MEKK (B) Raf and (C) ZIK genes, under salt stress conditions in *C. sinensis*. The relative expression of these genes were analysed in different stages by using GraphPad Prism 8 software. The level of expression was in transcript per million (TPM).

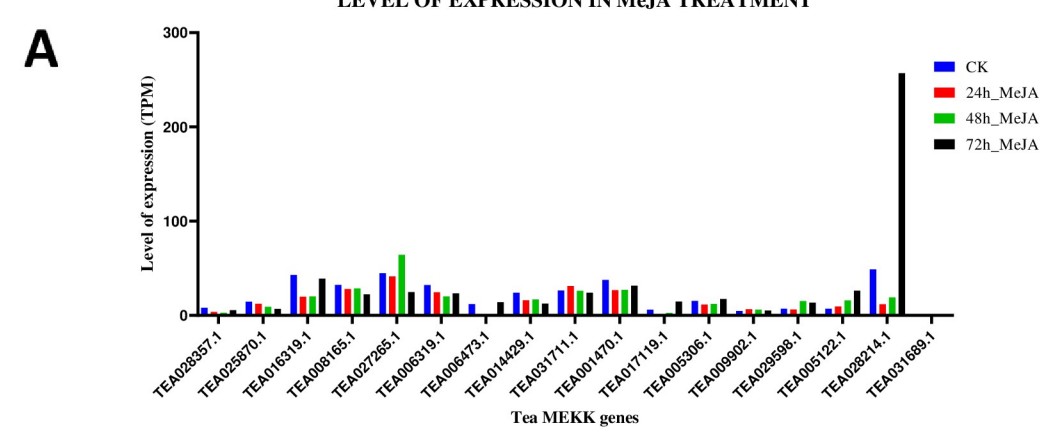

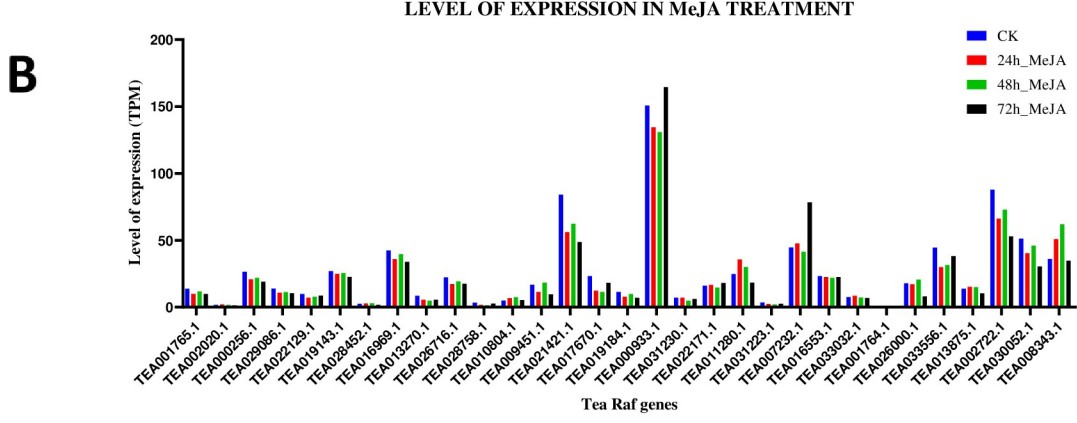

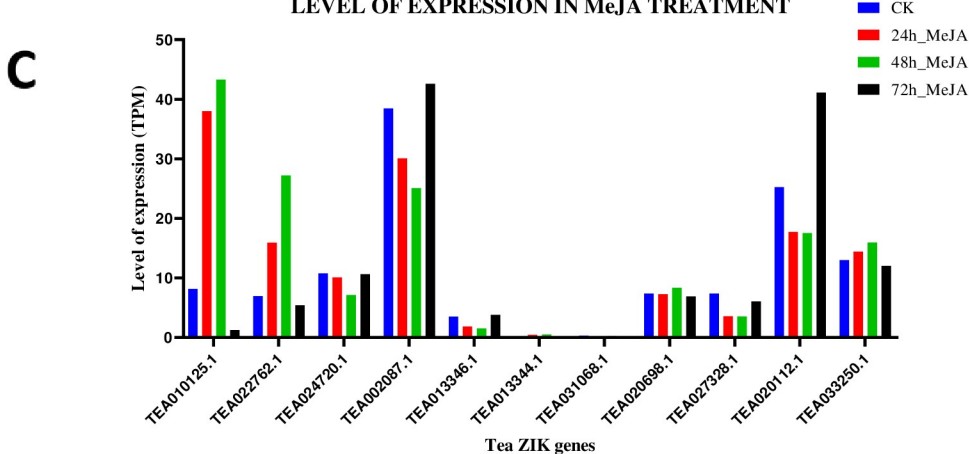

**Fig 12.** Gene expression patterns of (A) MEKK (B) Raf and (C) ZIK genes, under Methyl jasmonate (MeJA) treatment in *C. sinensis*. The relative expression of these genes were analysed in different stages by using GraphPad Prism 8 software. The level of expression was in transcript per million (TPM).

taken as the control. Analysis of the MEKK genes revealed that under NaCl-N-24h, 9 genes were upregulated and 8 genes were downregulated. For NaCl-N-48h condition, 9 genes showed upregulation and remaining 8 genes were downregulated. Expression levels under NaCl-N-72h revealed 5 genes being upregulated and the rest being downregulated (Fig 11A). For the Raf genes, expression data suggested that under NaCl-N-24h condition, 15 genes were upregulated and 16 genes were downregulated. Under the NaCl-N-48h condition, 16 genes showed upregulation and 15 genes were downregulated. Expression levels under NaCl-N-72h showed that 8 genes were upregulated and remaining 23 were downregulated (Fig 11B). For ZIK genes, 10 out of 11 genes had expression levels while 1 gene (TEA013344.1) had no effect under the given conditions. Expression levels under NaCl-N-24h condition, only 2 genes showed upregulation and the rest of the genes were downregulated. For NaCl-N-48h condition, only 1 gene was upregulated while the remaining 9 were downregulated. NaCl-N-72h condition too revealed a similar result with 2 genes being upregulated and remaining 8 being downregulated (Fig 11C). Heat maps were generated for the above-mentioned data as well (S12 Fig in S2 File).

Finally, the expression levels of the tea MAPKKKs under MeJA treatment were studied and analysed. The hormonal treatment data is recorded based on the results of exposing the plant parts to aqueous solution of MeJA, under 4 stages: 1. 0h: 2. 12h: 3. 24h and 4. 48h where, 0h was used as the control. For the MEKK genes, under the 12h_MeJA condition, 3 genes showed upregulation, 13 genes were downregulated and remaining 1 gene had no expression at all. Under the 24h_MeJA condition, 4 genes were upregulated, 12 were downregulated and 1 gene was not expressed at all. Under 48h_MeJA condition, 8 genes were upregulated and 9 genes were downregulated (Fig 12A). Similarly, for the Raf genes, treatment under 12h_MeJA condition revealed that 10 out of 31 genes were upregulated and remaining 21 genes were downregulated. Under 24h_MeJA condition, 8 genes showed upregulation while 23 genes were downregulated. 48h_MeJA revealed that only 4 genes were upregulated and rest of the genes were downregulated (Fig 12B). ZIK genes under the 12h_MeJA condition revealed that 4 genes were upregulated and 7 genes were downregulated. 24h_MeJA condition showed 5 genes being upregulated and remaining 6 being downregulated. 48h_MeJA condition suggested that 3 genes were upregulated and remaining 8 being downregulated (Fig 12C). Heat maps for these data were also generated (S13 Fig in S2 File). A similar approach was taken for highlighting biotic stress responsive and defensive role of chitinase genes in tea [51].

## Discussion

The MAPKKK-MAPKK-MAPK signalling cascade plays an important role in plant development as well as in response to various environmental stresses [5, 35, 52]. Investigation of the MAPKKK genes, which form a significant component of this core regulatory network would certainly aid to a better understanding of the signalling genes. Although much progress has been made in identifying the functions of MAPKKK genes in many organisms, these genes are yet to be analysed in *C. sinensis*. The objective of this study was to provide a comprehensive synopsis of the phylogenetic relationship, intron-exon architecture, motifs, functional domains, cis regulatory elements, genomic distribution and expression patterns of the MAPKKK genes in tea. Herein, a grand total of 59 MAPKKK proteins were screened and identified from tea plant genome. Previous studies in Arabidopsis [34], cucumber [10] and rice [6] have showed that the genes of MAPKKK family are classified into 3 subfamilies namely MEKK, Raf and ZIK [6, 10, 34]. Phylogenetic analysis (Fig 1) in tea showed similar results which indicate that genes in tea are also classified into these subfamilies. All the identified tea MAPKKKs had their respective subfamily specific domains. Motif analyses revealed that all

MAPKKK proteins had protein kinase domains and proteins belonging to the same subfamily shared similar motifs (Fig 3). This result is consistent to previous studies conducted on other plants like cucumber [10], Arabidopsis [34] and banana [53]. The study of intron-exon architecture in tea MAPKKK genes revealed a significant variation in the number of introns and exons (Fig 4). The average number of exons in MEKK genes ranged between 6 to16. Highest number of exons found among the MEKK genes was 18. Raf genes had an average of 6 to 18 exons, with the highest being a staggering 28 found in TEA016969.1 and ZIK genes had exons between 7 to 10. Raf subfamily thereby featured more number of introns than MEKK and ZIK subfamilies. Reports suggest that the rate at which introns are lost is faster compared to the rate at which introns are gained after segmental duplication [54]. This is a conclusive evidence to infer that Raf subfamily might contain the original set of genes, from which genes of other subfamilies have been derived. The analysis also proposed that genes belonging to the same subgroup featured similar intron-exon organization. The MAPKKK genes also displayed a significant variation with respect to the UTR segments. Most genes possessed both 5' and 3' UTRs while few had only the 5' UTR or 3' UTR segment. These variations of the gene structures suggest that the tea genome has been variable during its evolutionary history. Similar occurrence was also observed in plants like cassava [55], grapevine [56] and maize [40]. The interactions between the transcription factors and the promoter binding sites have crucial roles in regulation of gene expression at the transcriptional level [44]. The promoter sequence analysis for all the tea MAPKKKs revealed the diverse variety of cis-acting regulatory elements and their respective biological functions (Fig 5). Further in the study, all the identified genes were mapped onto their respective scaffolds (Fig 6). Duplication events observed among the ZIK genes shows the evidence that these genes play a crucial role in *C. sinensis*. The ratio of non-synonymous substitution rates ($K_a$) and synonymous substitution rates ($K_s$) was evaluated which indicated strong positive selection pressures to have occurred, enabling different factors to regulate the MAPKKKs in *C. sinensis* genome.

Earlier, comprehensive study in other plants has shown that MAPK cascade genes are extensively involved in controlling a number of biological processes which include cell growth, proliferation and response to various biotic and abiotic stresses such as salt stress, cold stress and drought stress [57–60]. Tea plant is a woody perennial tree and has a life span of more than 100 years. [61]. However, traditional breeding techniques for tea are slow and primarily limited to selection, which leads to narrowing down of its genetic base. Plants develop numerous signalling pathways to convert external stimuli into intracellular reactions in order to defend themselves against various environmental stress factors [62, 63]. MAPKKKs function at the highest level of the MAPK signalling cascade, helping with development and stress tolerance in plants.

A receptor mediated activation of MAPKKK proteins receive upstream signals to activate MAPKK proteins by phosphorylating the serine/threonine residues of the conserved motif (Ser/Thr-X3−5-Ser/Thr) [64, 65], which further phosphorylate specific MAPKs [66]. The activated MAPKK proteins further activates downstream MAPK proteins in the T-X-Y motif [64, 65]. The phosphorylated MAPK proteins then activate multiple downstream target proteins, including transcription factors, protein kinases, and cytoskeletal components [64, 65].

MAPKKKs have been extensively studied in Arabidopsis and have been well characterised. Previous literatures have conveyed that MEKK1-MKK1-MPK4 cascade is activated following a wounding stress response [67]. The MEKK1-MKK2-MPK4/MPK6 cascade is stimulated in salt and cold stress conditions [68]. Biochemical and genetic research suggests that MEKK1 is critically significant in response to cold stress and salt stress in Arabidopsis [69]. MAPK proteins classified in the same clades have been reported to perform similar roles in different organisms [70, 71]. Expression data presented in this study revealed TEA005122.1 had the

highest level of expression under salt stress and this gene belongs to clade A along with AtMEKK1. A similar clustering event with AtMEKK1 was observed for TEA006319.1, which displayed the highest levels of expression under cold stress. Hence, TEA005122.1 and TEA006319.1 might get activated in response to cold and salt stress and initiate MEKK1 signalling cascade in tea. MKK3 encodes for a Mitogen Activated Protein Kinase Kinase, that stimulates MPK8, and is a target of MPKKK20, regulating ROS accumulation. MKK3-MAPKKK17-MAPKKK18 form an element of the ABA signalling pathway. MAPKKK17 and MAPKKK18 belong to Ser/Thr protein kinase family and help in the ABA-dependent activation of the MKK3-MPK7 pathway [72]. Previous study has shown that abcissic acid mediates drought stress response [73]. In our study, TEA005122.1 belonging to clade A is homologous to AtMEKK18 and shows the highest level of expression under drought stress, which may be suggestive of the fact that TEA005122.1 might be responsible for drought stress responsive pathway in tea.

Analysis of gene expression in different plant parts under various environmental stress stimuli is key to understand the functions of the genes. Among the MEKK genes, TEA016319.1 was expressed consistently in all the 8 plant tissues (Fig 8A). While for the Raf and ZIK genes, TEA000933.1 and TEA002087.1 were the consistently expressed genes (Fig 8B and 8C). Reactive oxygen species (ROS) are oxygen derivatives, which are highly reactive by-products of the aerobic metabolism [74]. Plants consist of a complicated network of ROS scavenging antioxidant enzymes that helps to regulate the ROS levels under normal physiological conditions [74, 75]. Although a change from normal physiological conditions to adverse conditions shifts the equilibrium, resulting in increased ROS production. This might lead to serious oxidative damage and cell death because ROS are highly toxic to the cellular machinery [75]. Studies have suggested that the MAPK signalling cascade comprising of the MAPKKK-MAPKK-MAPK module is stimulated when excess ROS levels are detected under different stress conditions such as salt stress, cold stress and drought stress [74, 75]. It has been revealed that MPKKK1 activates two of its highly homologous MAPKKs (MKK1 and MKK2), which operate upstream of both MPK4 and MPK6 [68, 75]. Expression data for treatment under Methyl jasmonate (MeJA) revealed that TEA028214.1 among the MEKK genes, TEA000933.1 among the Raf genes and TEA002087.1 among the ZIK genes were expressed the most under the 72_MeJA condition (Fig 12). Collectively, these findings suggest the involvement of a number of MAPKKK genes, being upregulated and expressed under the stress conditions. In general, this study provides a detailed and comprehensive analysis of the MAPKKK genes in tea. Further extensive studies needs to be conducted on MAPKKK genes of tea that could provide a better understanding on the various functions of these set of genes in developmental processes and expression under various abiotic stress stimuli.

## Conclusion

Mitogen activated protein kinases (MAPK) signalling cascade plays significant roles in different biological processes. The signalling components are linked to the upstream and downstream regulators by phosphorylation. There has been substantial development in identifying the different MAPKKK genes and understand their physiological roles in various plants. However, these genes had not been yet explored and studied in tea plant. *In-silico* genome wide analysis had identified 59 MAPKKK genes from *C. sinensis* genome. The classification of the identified MAPKKK genes in 3 subfamilies were conducted based on their specific domain signatures. The genes were further investigated under phylogenetic analysis, conserved protein motifs, intron-exon architecture and analysis of cis regulatory elements. The 59 genes were mapped onto their respective genomic scaffolds and a network of functionally interacting

genes was constructed. Further, expression profile analyses were conducted to reveal the involvement of the tea MAPKKK genes in various tissues during development and understand the expression pattern of these genes under various abiotic stress stimuli and plant hormonal treatment. These data will provide detailed information about the tea MAPKKK genes for further characterization of the MAPK signalling cascade and lay a concrete foothold for further exploration and research on *C. sinensis.*

## Supporting information

**S1 File. This file contains all the supporting tables.**
(PDF)

**S2 File. This file contains all the supporting figures.**
(PDF)

## Acknowledgments

The authors are thankful to DBT-eLibrary Consortium (DeLCON) for providing access to e-resources.

**Author details**: Abhirup Paul: Department of Biochemistry, REVA University, Bangalore, Karnataka, India (Email: abhirupm16@gmail.com); Anurag P. Srivastava: Department of Life Sciences, Garden City University, Bangalore, Karnataka, India (Email: anurag.srivastava@gardencity.university, anuiitkgp@gmail.com); Shreya Subrahmanya: Department of Botany, St. Joseph's college autonomous, Bengaluru, Karnataka, India (Email: shreyasub916@gmail.com); Guoxin Shen: Sericultural Research Institute, Zhejiang Academy of Agricultural Sciences, Hangzhou 310021, China (Email: guoxin.shen@ttu.edu); Neelam Mishra: Department of Botany, St. Joseph's college autonomous, Bengaluru, Karnataka, India (Email: neelamiitkgp@gmail.com, neelammishra@sjc.ac.in).

## Author Contributions

**Formal analysis:** Neelam Mishra.

**Funding acquisition:** Guoxin Shen.

**Investigation:** Anurag P. Srivastava, Guoxin Shen.

**Methodology:** Abhirup Paul, Anurag P. Srivastava, Shreya Subrahmanya, Neelam Mishra.

**Resources:** Neelam Mishra.

**Software:** Abhirup Paul.

**Supervision:** Guoxin Shen, Neelam Mishra.

**Validation:** Abhirup Paul, Guoxin Shen.

**Writing – original draft:** Abhirup Paul, Neelam Mishra.

**Writing – review & editing:** Neelam Mishra.

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
