## [Decision Letter · Decision Letter 0]

14 Apr 2021

PONE-D-21-05595

In-silico identification, expressional profile and regulatory network analysis of Mitogen Activated Protein Kinase Kinase Kinase gene family in C. sinensis

PLOS ONE

Dear Dr. Mishra,

Thank you for submitting your manuscript to PLOS ONE. After careful consideration, we feel that it has merit but does not fully meet PLOS ONE’s publication criteria as it currently stands. Therefore, we invite you to submit a revised version of the manuscript that addresses the points raised during the review process.

We look forward to receiving your revised manuscript.

Kind regards,

Tapan Kumar Mondal, Ph.D

Academic Editor

PLOS ONE

Journal Requirements:

2. Please include captions for your Supporting Information files at the end of your manuscript, and update any in-text citations to match accordingly. Please see our Supporting Information.

Additional Editor Comments (if provided):

The work is good but needs improvement. I am agree with the reviewers about the issues raised by them. In addition I suggest to add the classical and related paper on tea biotechnology which are missing:-

1. Mukhopadhyay et al (2016) Plant Cell Rep 35(2):255-87.

2. Bordoloi et al (2021). Physiol Mol Biol Plants 27, 369–385

Reviewers' comments:

Reviewer's Responses to Questions

**Comments to the Author**

1. Is the manuscript technically sound, and do the data support the conclusions?

Reviewer #1: Yes

Reviewer #2: Partly

2. Has the statistical analysis been performed appropriately and rigorously? 

Reviewer #1: N/A

Reviewer #2: N/A

3. Have the authors made all data underlying the findings in their manuscript fully available?

Reviewer #1: Yes

Reviewer #2: No

4. Is the manuscript presented in an intelligible fashion and written in standard English?

Reviewer #1: Yes

Reviewer #2: Yes

5. Review Comments to the Author

Reviewer #1: Comments to the Authors

Please find the attachments (pdf version of manuscript & supplementary files) for details.

In this work, authors have carried out genome-wide In-silico identification, expressional profile and regulatory network analysis of Mitogen Activated Protein Kinase Kinase Kinase gene family in C. sinensis by retrieval of genome, proteome & gene expression data from TPIA database. There is no error in terms of the technical aspects of the work done through bioinformatics approach and the information generated is presented in an appropriate way. Though, the manuscript is well written, there are some minor corrections and clarifications needs to be made before it is accepted for publication that will help to improve the readability & presentation of the work. Further, authors need to do an additional work which is very common part of genome wide In-silico mining of gene families. The authors need to retrieve the genomic DNA sequences upstream of the transcriptional start site for each tea MAPKKK gene from TPIA database for identifying putative promoter cis-acting elements. For identifying cis elements, the authors may use PlantCARE database (http://bioinformatics.psb.ugent.be/webtools/plantcare/html/).

All the corrections needed to be done are highlighted and commented in the pdf version of the manuscript & supplementary files provided. The questions pertaining to the text in the manuscript which needs to be answered are also provided as comments in the manuscript pdf version. Please find the attached pdf version of manuscript & supplementary files for comments. Few points where correction & revision (Minor revision) is required are given below & commented in details in the pdf version of the manuscript and supplementary files.

1. Few grammatical and typographical errors needs to be corrected.

2. All the abbreviated forms are required to be written in their full form at the first instance of their occurrence.

3. Reframing of few sentences is required.

4. Few questions related to homologous/orthologous gene pairs and few others questions needed to be answered.

5. For some statements references need to be cited.

6. Some typographical error regarding the total number of plant tissue samples whose expression have been analysed need to be corrected.

7. Some figures caption/legend description needs to be corrected.

8. Specific tissue names whose expression was analysed under different abiotic stress treatment & MeJA treatment needs to be mentioned.

9. Throughout the discussion in many instances some sentences almost seemed to be repetitive or similar to what was already mentioned in introduction and result section. Authors are advised to cross check this and modify or reframe the sentences wherever needed.

10. Scientific names need to be italicized in the Supplementary Tables captions. For Example- C.sinensis.

11. In the supplementary figures: the heatmap labels, the numbers at the bottom and alphabets on the right side, each needs to be described specifically what they correspond to.

Major Revision

Additional Work: The authors need to retrieve the genomic DNA sequences upstream of the transcriptional start site for each tea MAPKKK gene from TPIA database for identifying putative promoter cis-acting elements. For identifying cis elements, the authors may use PlantCARE database (http://bioinformatics.psb.ugent.be/webtools/plantcare/html/).

Reviewer #2: 1) Few MAPKKK genes identified In-silico genome wide analysis from C. sinensis genome in this study should be validated by wet lab.

2) The identified MAPKKK genes in different subfamilies should not just be based on their phylogenetic relationships.

3) The network of functionally interacting genes should also include the genes that have experimentally validated support.

4) Expression profile analysis by qRT-PCR is required to reveal the involvement of the tea MAPKKK genes in various tissues during development and under various abiotic stress stimuli and plant hormonal treatment.

5) The study should decipher a model of signalling mechanism mediated by MAPKKK genes cascade in tea.

6. PLOS authors have the option to publish the peer review history of their article (what does this mean?). If published, this will include your full peer review and any attached files.

Reviewer #1: **Yes: **Madhurjya Gogoi

Reviewer #2: No

---

## [Author Response · Author response to Decision Letter 0]

5 Jun 2021

Dear Editor, 

Two references as mentioned by you are now added in the revised manuscript.

Reviewer 1

Minor Comments

1. Few grammatical and typographical errors need to be corrected.

>Response: The grammatical and typographical errors have been corrected and are highlighted as yellow in the revised manuscript.

2. All the abbreviated forms are required to be written in their full form at the first instance of their occurrence.

>Response: All the abbreviated forms are written in their full form at the first instance of their occurrence in the revised manuscript.

3. Reframing of few sentences is required.

>Response: The sentences have been reframed in the revised manuscript under the yellow highlights as suggested by the reviewer.

4. Few questions related to homologous/orthologous gene pairs and few others questions needed to be answered.

>Response: We agree with the reviewer that few questions related to homologous/orthologous gene pairs and few other questions needs to be addressed in the manuscript. We have provided a detailed response of this question (Manuscript comments A and B) and have incorporated the corresponding changes in the revised manuscript (page 12). 

Manuscript comments

A. What does authors mean here by saying Raf tree did not feature any orthologous gene pair? Does authors mean to say that Raf phylogenetic tree did not contain any tea orthologous gene from arabidopsis or any other species? If they are not homologous or orthologous genes how they are clustered in the same clade of the Raf phylogenetic tree? Please clarify a bit.

B. What does authors mean here by saying ZIK tree did not feature any orthologous gene pair? Does authors mean to say that ZIK phylogenetic tree did not contain any tea orthologous gene from arabidopsis or any other species? If they are not homologous or orthologous genes how they are clustered in the same clade of the ZIK phylogenetic tree? Please clarify a bit.

>Response: We thank the reviewer for the comment. The phylogenetic analysis of the Raf and ZIK genes of tea showed that the respective genes have clustered together in the same clade, along with the other Raf and ZIK genes identified from different plant species. This is suggestive of the fact that the genes are either homologous or orthologous to each other. However, the Raf and ZIK genes did not feature any orthologous gene with respect to Arabidopsis. This was validated only when the Raf and ZIK gene accession IDs were scanned in the TPIA database to search for the presence of orthologous genes in the later part of the study (Functional Interaction Network). 

The reason for clustering of the Raf and ZIK genes under the same clade in their respective phylogenetic tree despite not being orthologous is that; orthologs are defined as genes that share a common ancestor by speciation which led to the genes clustering together. So, the identification of orthologous genes via in-silico approach is purely based on the assumption that these orthologous genes start to diverge after a speciation event and may diverge within the same clade or an adjacent clade, sharing the common ancestor (Wu et al., 2006). It is thus possible for such orthologous groups to exist in a phylogenetic tree. But, when these Raf and ZIK genes were further searched in the TPIA database, it showed the absence of any orthologous genes. 

The same thing has been clarified and the corresponding changes have been made in the revised manuscript under the heading “Phylogenetic analysis of tea MAPKKKs” in the “Results” section with a line beginning with “The results are suggestive of the fact……….” (page 12).

Reference: Wu, Feinan et al. “Combining bioinformatics and phylogenetics to identify large sets of single-copy orthologous genes (COSII) for comparative, evolutionary and systematic studies: a test case in the euasterid plant clade.” Genetics vol. 174,3 (2006): 1407-20. doi:10.1534/genetics.106.062455

C. Authors have mentioned the function of tea orthologues in Arabidopsis that formed a part of the interaction network. However, they did not cite any reference on the basis of which they have included these statements. If they have taken from STRING server or database, they need to mention the same or if they have taken from literature then cite the reference.

>Response: The function of tea orthologues in Arabidopsis that formed a part of the interaction network was taken from TAIR database and the reference for the same has been provided in the revised manuscript under the heading “GO ontology analysis and functional interaction network of tea MAPKKKs” in the “Results” section (page 19). 

D. Authors have mentioned that they have extracted expression data pertaining to MAPKKKs genes from TPIA database under different abiotic stress condition. They need to mention here about the specific tissue or organ of tea plant to which these expression data correspond. Authors are advised to look for this information in the TPIA database or any of the published research papers on this and incorporate the information here.

>Response: We thank the reviewer for the comment. The expression data of the leaves of the tea plant present in TPIA database was used to study the effect of different abiotic stress condition and the same has also been mentioned in the revised manuscript in the “Results” section under the heading “Abiotic stress induced differential expression levels of tea MAPKKKs” with a line beginning with “The expression data of the leaves of the tea plant………..” (page 21).

5. For some statements references need to be cited.

>Response: The references for the statements as mentioned by the reviewer have been cited in the revised manuscript.

6. Some typographical error regarding the total number of plant tissue samples whose expression have been analysed need to be corrected.

>Response: We thank reviewer for pointing this. We have corrected the results in the revised manuscript. There are eight genes in total and results have been provided appropriately (page 20-21).

7. Some figures caption/legend description needs to be corrected.

>Response: The figure legends as pointed out by the reviewer have been corrected in the revised manuscripts as well as in the supplemental files. 

8. Specific tissue names whose expression was analysed under different abiotic stress treatment & MeJA treatment needs to be mentioned.

>Response: The expression analysis under different abiotic stress treatment and MeJA treatment was done for the leaves of the tea plant and the same has been mentioned in the revised manuscript.

9. Throughout the discussion in many instances some sentences almost seemed to be repetitive or similar to what was already mentioned in introduction and result section. Authors are advised to cross check this and modify or reframe the sentences wherever needed.

>Response: The repetitive statements in the discussion section have been modified in the revised manuscript.

10. Scientific names need to be italicized in the Supplementary Tables captions. For Example- C. sinensis.

>Response: The scientific names in the Supplementary Tables captions have been italicized in the revised manuscript.

11. In the supplementary figures: the heatmap labels, the numbers at the bottom and alphabets on the right side, each needs to be described specifically what they correspond to.

>Response: The numbers at the bottom corresponds to the tea genes while the alphabets on the right side represent 8 different tissues. The name of the tea genes and the name of the 8 different tissues have been provided in “S9 Fig.” of Supplemental information 2. 

 Similarly, for abiotic stress the numbers at the bottom corresponds to the tea genes while the alphabets on the right side represent different experimental stages. The name of the tea genes and different stages has been provided in “S10-S13 Figs.” of Supplemental information 2.

Major revision 

1. Additional Work: The authors need to retrieve the genomic DNA sequences upstream of the transcriptional start site for each tea MAPKKK gene from TPIA database for identifying putative promoter cis-acting elements. For identifying cis elements, the authors may use PlantCARE database (http://bioinformatics.psb.ugent.be/webtools/plantcare/html/).

>Response: We thank the reviewer for this comment. The genomic DNA sequences upstream of the transcriptional start site for each tea MAPKKK gene has been retrieved from TPIA database in order to identify putative promoter cis-acting elements in the methodology and the results of these experiment has been provided in the revised manuscripts under the heading “Analysis of cis-regulatory elements” and “Retrieval of promoter sequences and analysis of cis-regulatory elements” (page 6 and page 15) respectively.

Reviewer 2

Major Comments

1. Few MAPKKK genes identified In-silico genome wide analysis from C. sinensis genome in this study should be validated by wet lab. 

>Response: The main aim of this study was to identify the MAPKKK genes in tea using an in-silico approach. Expression analysis data presented in this study have already been experimentally verified by other researchers and is available in the TPIA database. In addition, all the identified genes were subjected to GO ontology analysis in order to predict the potential functions of all the 59 tea MAPKKKs. The corresponding text for the same has been added in the revised manuscript under the heading “GO ontology analysis and functional interaction network of tea MAPKKKs” with a line beginning with “The GO ontology analysis was performed in-order to predict the potential functions” (page 18). 

2. The identified MAPKKK genes in different subfamilies should not just be based on their phylogenetic relationships. 

>Response: The classification of the identified MAPKKK genes in 3 subfamilies were conducted based on their specific domain signatures (“Domain analysis of tea MAPKKKs”, page 12). The genes were further investigated under phylogenetic analysis, conserved protein motifs, intron-exon architecture and analysis of cis regulatory elements. After performing all of these analyses, the MAPKKK genes were classified into the three different subfamilies namely MEKK, ZIK and Raf sub-family. The corresponding text for the same has been added in the revised manuscript under the “Conclusion” section with a line beginning with “The classification of the identified MAPKKK genes……..” (page 30). 

3. The network of functionally interacting genes should also include the genes that have experimentally validated support.

>Response: The genes included in the functional interaction network present in this study have been taken from the TAIR database, which are already experimentally verified. Further, the functions of the tea genes present in the network were validated by the TPIA database and GO ontology. The references for the same has been added and the text has been modified accordingly in the revised manuscript under the heading “GO ontology analysis and functional interaction network of tea MAPKKKs” (page 18-20).

4. Expression profile analysis by qRT-PCR is required to reveal the involvement of the tea MAPKKK genes in various tissues during development and under various abiotic stress stimuli and plant hormonal treatment.

>Response: TPIA database has the expression profile data of all the C. sinensis genes, which have been already experimentally verified. The expression profile data of the identified MAPKKK genes presented in this study have been retrieved from the TPIA database. 

References:

a) For cold stress:

Wang X, Zhao Q, Ma C, et al. Global transcriptome profiles of Camellia sinensis during cold acclimation. BMC Genomics. 2013;14:415.

b) For salt and drought stress:

Zhang Q, Cai M, Yu X, et al. Transcriptome dynamics of Camellia sinensis in response to continuous salinity and drought stress. Tree Genet Genomes. 2017;13:78

c) For MeJA treatment:

Shi, J., Ma, C., Qi, D., Lv, H., Yang, T., Peng, Q., Chen, Z. et al. (2015) Transcriptional responses and flavor volatiles biosynthesis in methyl jasmonate-treated tea leaves. BMC Plant Biol. 15, 233

d) For tissue specific:

Wei, C.L., Yang, H., Wang, S.B., Zhao, J., Liu, C., Gao, L.P., Xia, E.H. et al. (2018) Draft genome sequence of Camellia sinensis var. sinensis provides insights into the evolution of the tea genome and tea quality. Proc. Natl Acad. Sci. 115, E4151–E4158. 

5. The study should decipher a model of signalling mechanism mediated by MAPKKK genes cascade in tea.

>Response: A putative signalling mechanism mediated by MAPKKK genes cascade in tea has been included in the “Discussion” section of the revised manuscript.

---

## [Decision Letter · Decision Letter 1]

11 Aug 2021

PONE-D-21-05595R1

In-silico identification, expressional profile and regulatory network analysis of Mitogen Activated Protein Kinase Kinase Kinase gene family in C. sinensis

PLOS ONE

Dear Dr. Mishra,

Thank you for submitting your manuscript to PLOS ONE. After careful consideration, we feel that it has merit but does not fully meet PLOS ONE’s publication criteria as it currently stands. Therefore, we invite you to submit a revised version of the manuscript that addresses the points raised during the review process. Especially, queries raised by the Reviewer#3 on the title and statistics need to be addressed.

We look forward to receiving your revised manuscript.

Kind regards,

Ramegowda Venkategowda, PhD

Academic Editor

PLOS ONE

Journal Requirements:

Reviewers' comments:

Reviewer's Responses to Questions

**Comments to the Author**

1. If the authors have adequately addressed your comments raised in a previous round of review and you feel that this manuscript is now acceptable for publication, you may indicate that here to bypass the “Comments to the Author” section, enter your conflict of interest statement in the “Confidential to Editor” section, and submit your "Accept" recommendation.

Reviewer #1: (No Response)

Reviewer #3: (No Response)

2. Is the manuscript technically sound, and do the data support the conclusions?

Reviewer #1: Yes

Reviewer #3: Yes

3. Has the statistical analysis been performed appropriately and rigorously? 

Reviewer #1: N/A

Reviewer #3: No

4. Have the authors made all data underlying the findings in their manuscript fully available?

Reviewer #1: Yes

Reviewer #3: Yes

5. Is the manuscript presented in an intelligible fashion and written in standard English?

Reviewer #1: Yes

Reviewer #3: Yes

6. Review Comments to the Author

Reviewer #1: The authors have substantially improved the revised manuscript and incorporated all the necessary changes required. I am satisfied with the authors response on the queries asked. However, before it is accepted for publication, I would like to know a bit about the process of orthologous gene identification authors have employed for my clarification.

• Did the authors use all the Arabidopsis MEKK, Raf and ZIK genes accession one by one and looked for the orthologs in C.sinenis TPIA database Orthologous Groups search option?

• Does TPIA database Orthologous Groups search option includes all the C.sinensis gene accessions that are orthologous to other species?

For future similar kind of work, I would advise authors to incorporate synteny analysis between the organism of interest and a model or reference organism. It is not required for this paper now. Authors just needs to address the concern I have raised above regarding orthologous gene identification. Further, authors need to make few minor changes in the manuscript before acceptance.

(1) I have strikeout and underlined few words in some sentences and also provided comments on those within the revised manuscript pdf file (revised manuscript with track changes part of the pdf file). Please find the attached file. Authors are advised to incorporate the required changes.

(2) The authors need to correct the legends of Fig 9, Fig 10, Fig 11, Fig 12.

The relative expression of the genes were analysed in the leaves of C.sinensis in different stages and not in different tissues for the Fig 9, Fig 10, Fig 11, Fig 12. Authors wrote “The relative expression of these genes were analysed in different tissues by using GraphPad Prism 8 software”.

Authors need to incorporated the corrected legend for Fig 9, Fig 10, Fig 11, Fig 12. The corrected legend for Fig 9, Fig 10, Fig 11, Fig 12 are given below:

Fig 9. Gene expression patterns of (A) MEKK (B) Raf and (C) ZIK genes, under cold

stress conditions in C. sinensis. The relative expression of these genes were analysed in

different stages by using GraphPad Prism 8 software. The level of expression was in

transcript per million (TPM).

Fig 10. Gene expression patterns of (A) MEKK (B) Raf and (C) ZIK genes, under

drought stress conditions in C. sinensis. The relative expression of these genes were

analysed in different stages by using GraphPad Prism 8 software. The level of expression

was in transcript per million (TPM).

Fig 11. Gene expression patterns of (A) MEKK (B) Raf and (C) ZIK genes, under salt

stress conditions in C. sinensis. The relative expression of these genes were analysed in different stages by using GraphPad Prism 8 software. The level of expression was in

transcript per million (TPM).

Fig 12. Gene expression patterns of (A) MEKK (B) Raf and (C) ZIK genes, under

Methyl jasmonate (MeJA) treatment in C. sinensis. The relative expression of these genes

were analysed in different stages by using GraphPad Prism 8 software. The level of

expression was in transcript per million (TPM)

Reviewer #3: The authors of this study used an insilico approach for phylogenetic, structural and functional characterization of the MAPKKK gene family in tea. The authors use standard bioinformatics methods for the analysis and the results seem to match expectations. While most of these methods are well-described in the revised version, I have a few additional concerns that must be addressed before publication.

Major:

-The title is suggestive of a rigorous regulatory network analysis of MAPKKK in tea. However, this does not seem to be the case here. In its current form, this network analysis pipeline makes no sense because no biological knowledge about tea kinases is extracted and highlighted anywhere in the MS. Furthermore, apart from the title, several other parts of the MS are misleading. In methods, the network analysis section suggests that a network of all orthologs between tea and arabidopsis was built, but in results it seems only one gene was used as a guide.

1) Explain why only one gene and why that one?

2) I am curious to know how many of the original 59 genes (also mentioned in the abstract) are part of the network shown?. The authors can also show the overlaps, if any, between the interacting partners of all tea kinases that match Arabidopsis kinases.

3) How does the network shown tell us about the predicted kinases?

In my opinion, a better network analysis strategy could be employed, or the title should be modified to reflect other important analysis performed (e.g. phylogenetic analysis).

-The authors mention that they used GO analysis for predicting gene function. However, the authors did not perform any statistical tests to support the results. GO analysis is not really a prediction method, but in this case, it can be used to validate one’s expectation from the nature of the study (e.g. terms like ‘protein phosphorylation’).

Please describe the following points in greater detail:

1) Exactly how was the GO analysis performed? The link provided does not work. Did you use sequences (Blast2GO?), gene IDs, or ortholog gene IDs as input to topGO?

2) Is the overlap between tea genes with GO terms shown in Fig. S8 statistically significant? Same goes for CC and MF analysis.

Minor:

1) Figure legends should not be repetitions of the methods. Please clearly explain in the legend what is shown in the plot, specially in the supplemental figures Fig. S1-3. You should not expect a general reader to already know the outputs of TMHMM Server. What do probabilities above 1 mean? What are the red peaks?

2) The tradition set by the original creators of GO is to refer to them as ‘ontologies’ and ‘terms’, rather than ‘groups’ and ‘subgroups’, respectively. Please rephrase to stick with the convention.

3) In the abstract, the sentence “..on the basis of orthologous genes in Arabidopsis, functional interaction was carried out in C. sinensis.” is suggestive of wet-lab experiments. Please rephrase to clearly reflect network analysis of one of the 59 genes the story revolves around.

4) Page 19: I believe that the lines between “Similarity search programs like BLAST…...tend to possess similar functions” are to provide a logical reasoning for your approach of using Arabidopsis network in STRING. If this is the case, please rephrase these lines to reflect the exact logic. It’s somewhat confusing in its current form.

7. PLOS authors have the option to publish the peer review history of their article (what does this mean?). If published, this will include your full peer review and any attached files.

Reviewer #1: **Yes: **Madhurjya Gogoi

Reviewer #3: **Yes: **Chirag Gupta

---

## [Author Response · Author response to Decision Letter 1]

21 Sep 2021

# Reviewer 1

The authors have substantially improved the revised manuscript and incorporated all the necessary changes required. I am satisfied with the authors response on the queries asked. However, before it is accepted for publication, I would like to know a bit about the process of orthologous gene identification authors have employed for my clarification.

1. Did the authors use all the Arabidopsis MEKK, Raf and ZIK genes accession one by one and looked for the orthologs in C. sinenis TPIA database Orthologous Groups search option?

>Response: Yes, all the identified tea MAPKKKs (17 MEKK, 31 Raf and 11 ZIK genes) were searched one by one in the TPIA database Orthologous Groups search option (Orthologous Groups (shengxin.ren)) to find the presence any orthologous genes between tea and Arabidopsis. The same thing has also been mentioned in the revised manuscript under the “GO ontology analysis and functional interaction network of tea MAPKKKs” in the “RESULTS” section with a line beginning with “All the identified tea MAPKKKs……….” (Page 19).

2. Does TPIA database Orthologous Groups search option includes all the C. sinensis gene accessions that are orthologous to other species?

>Response: Yes, the TPIA database orthologous group search option includes the list of all the C. sinensis genes which are orthologous to 11 representative plant species, namely Kiwi (Actinidia chinensis), Amborella (Amborella trichopoda), Arabidopsis (Arabidopsis thaliana), Coffee (Coffee Arabica), African oil palm (Elaeis guineensis), Medicago (Medicago truncatula), Poplar (Populus trichocarpa), Peach (Prunus persica), Cocoa (Theobroma cacao), Assam tea (Camellia teatre) and Grape (Vitis vinifera). The list for the same can also be found in TPIA database (Orthologous Groups (shengxin.ren)). This has been clarified in the revised manuscript under the section under the “GO ontology annotation and functional interaction network” in the “MATERIALS AND METHODS” section (Page 7).

Minor Comments

1. I have strikeout and underlined few words in some sentences and also provided comments on those within the revised manuscript pdf file (revised manuscript with track changes part of the pdf file). Please find the attached file. Authors are advised to incorporate the required changes.

>Response: All the required changes as suggested by the reviewer have been incorporated in the revised manuscript. 

2. The authors need to correct the legends of Fig 9, Fig 10, Fig 11, Fig 12.

>Response: We thank reviewer for the comment. The figure legends of the aforementioned figures (Fig 9, Fig 10, Fig 11, Fig 12) have been corrected in the revised manuscript. 

# Reviewer 2

Major Comments

1. The title is suggestive of a rigorous regulatory network analysis of MAPKKK in tea. However, this does not seem to be the case here. In its current form, this network analysis pipeline makes no sense because no biological knowledge about tea kinases is extracted and highlighted anywhere in the MS. Furthermore, apart from the title, several other parts of the MS are misleading. In methods, the network analysis section suggests that a network of all orthologs between tea and arabidopsis was built, but in results it seems only one gene was used as a guide.

i. Explain why only one gene and why that one?

>Response: In this study, the STRING server was used for constructing the functional interaction network in tea. Due to the absence of tea database in the STRING server, the Arabidopsis association model had to be employed to construct the functional interaction network (Wang et al., 2018, Chatterjee et al., 2020). In addition, studies have also shown that proteins sharing higher degree of sequence and structural similarities often tend to have similar functions (Gan et al., 2002; Wang et al., 2018). Among the 59 MAPKKKs in tea, only TEA005306.1 was found to be orthologous to AT5G55100.1. As no other genes were found to be orthologous to any other Arabidopsis MAPKKKS, AT5G55100.1 was used to build the interaction network. The interaction network thus generated helps us to predict how TEA005306.1 functionally interacts with other tea genes (Lui et al., 2021, Wang et al., 2018, Chatterjee et al., 2020)

References: 

1. Wang, YX., Liu, ZW., Wu, ZJ. et al. Genome-wide identification and expression analysis of GRAS family transcription factors in tea plant (Camellia sinensis). Sci Rep 8, 3949 (2018).

2. Chatterjee, A., Paul, A., Unnati, G.M. et al. MAPK cascade gene family in Camellia sinensis: In-silico identification, expression profiles and regulatory network analysis. BMC Genomics 21, 613 (2020).

3. Gan HH, Perlow RA, Roy S, Ko J, Wu M, Huang J, et al. Analysis of protein 844 sequence/structure similarity relationships. Biophys J. 2002;83(5):2781–91.

4. Liu, Z.; An, C.; Zhao, Y.; Xiao, Y.; Bao, L.; Gong, C.; Gao, Y. Genome-Wide Identification and Characterization of the CsFHY3/FAR1 Gene Family and Expression Analysis under Biotic and Abiotic Stresses in Tea Plants (Camellia sinensis). Plants 2021, 10, 570.

ii. I am curious to know how many of the original 59 genes (also mentioned in the abstract) are part of the network shown?. The authors can also show the overlaps, if any, between the interacting partners of all tea kinases that match Arabidopsis kinases.

>Response: The given interaction network shows 1 gene (TEA005306.1) out of all the 59 predicted MAPKKKs in tea, around which the network is constructed. The same has been mentioned in the revised manuscript under “GO ontology analysis and functional interaction network of tea MAPKKKs” in the “RESULTS” section.

 The given interaction network does not include any overlapping partners between the tea kinases and Arabidopsis kinases.

iii. How does the network shown tell us about the predicted kinases?

>Response: STRING is a database of known and predicted protein-protein interactions. The interactions include direct (physical) and indirect (functional) associations (https://string-db.org/). The main aim of the interaction network analysis was to see how the predicted kinases interact with the other genes in the tea genome (Wang et al., 2018; Chatterjee et al., 2020; Liu et al., 2021). Out of the predicted tea kinases, TEA005306.1 was found to be orthologous to one Arabidopsis kinase (AT5G55100.1), hence only this gene was used for generating the interaction network (Chatterjee et al., 2020). The interaction network thus given in the manuscript is to show how TEA005306.1 functionally interacts with other genes in the tea genome (Wang et al., 2018; Chatterjee et al., 2020; Liu et al., 2021). The text for the same has also been added in the revised manuscript under 

“GO ontology analysis and functional interaction network of tea MAPKKKs” in the “RESULTS” section with a line beginning with “This was done to predict……” (Page 19).

References: 

1. Wang, YX., Liu, ZW., Wu, ZJ. et al. Genome-wide identification and expression analysis of GRAS family transcription factors in tea plant (Camellia sinensis). Sci Rep 8, 3949 (2018).

2. Chatterjee, A., Paul, A., Unnati, G.M. et al. MAPK cascade gene family in Camellia sinensis: In-silico identification, expression profiles and regulatory network analysis. BMC Genomics 21, 613 (2020).

3. Liu, Z.; An, C.; Zhao, Y.; Xiao, Y.; Bao, L.; Gong, C.; Gao, Y. Genome-Wide Identification and Characterization of the CsFHY3/FAR1 Gene Family and Expression Analysis under Biotic and Abiotic Stresses in Tea Plants (Camellia sinensis). Plants 2021, 10, 570.

iv. In my opinion, a better network analysis strategy could be employed, or the title should be modified to reflect other important analysis performed (e.g. phylogenetic analysis).

>Response: We thank the reviewer for his suggestion. We agree that the title of the manuscript needs to be modified to reflect other important analyses performed in this study and so the title has been changed to, “In-silico genome wide analysis of Mitogen Activated Protein Kinase Kinase Kinase gene family in C. sinensis” in the revised version as suggested by the reviewer. 

2. The authors mention that they used GO analysis for predicting gene function. However, the authors did not perform any statistical tests to support the results. GO analysis is not really a prediction method, but in this case, it can be used to validate one’s expectation from the nature of the study (e.g. terms like ‘protein phosphorylation’). Please describe the following points in greater detail:

i. Exactly how was the GO analysis performed? The link provided does not work. Did you use sequences (Blast2GO?), gene IDs, or ortholog gene IDs as input to topGO?

>Response: We thank the reviewer for the comments. All the comments raised by the reviewer have been answered individually as given below.

Exactly how was the GO analysis performed?

>Response: The TPIA database provides the set of all the GO terms to which a particular tea gene is annotated (TPIA database > Tea gene annotation > Gene ontology classification). All the identified tea MAPKKKs were then searched individually to retrieve data of all the GO terms to which they have been annotated. The terms that appear for each searched gene, open up to AmiGO 2 server (AmiGO 2: Welcome (geneontology.org)), displaying detailed information for the respective GO terms. This data was reconfirmed by searching the GO terms in QuickGO server (https://www.ebi.ac.uk/QuickGO/). The obtained GO terms were then divided into 3 main categories i.e, Biological process (BP), Cellular component (CC) and Molecular function (MF) and a graph for the same was generated. Texts for the same has been mentioned in the revised manuscript under the “GO ontology annotation and functional interaction network” in the “MATERIALS AND METHODS” section (Page 7). 

The link provided does not work 

>Response: We thank reviewer for the comment. The link has been updated in the revised manuscript (https://www.ebi.ac.uk/QuickGO/) under the “GO ontology annotation and functional interaction network” in the “MATERIALS AND METHODS” section (Page 7). 

Did you use sequences (Blast2GO?), gene IDs, or ortholog gene IDs as input to topGO? 

>Response: The tea MAPKKKs genes were not used as input to topGO. The GO analysis was done using the TPIA database and AmiGO 2 server as described in the response to the first comment.

ii. Is the overlap between tea genes with GO terms shown in Fig. S8 statistically significant? Same goes for CC and MF analysis.

>Response: We thank reviewer for the comment. In order to find the statistical significance of the obtained GO terms, we have conducted a statistical analysis using TeaCoN database (Zhang et al., 2020). A GO statistics table has been added in Supporting Information File 1, titled “S8 Table: The GO enrichment analysis of all the 59 MAPKKKs of C. sinensis” and the GO enrichment figure has also been modified (S8 Fig.) in the revised Supporting Information files. 

Reference:

1. Rui Zhang, Yong Ma, Xiaoyi Hu, Ying Chen, Xiaolong He, Ping Wang, Qi Chen, Chi-Tang Ho, Xiaochun Wan, Youhua Zhang, and Shihua Zhang, TeaCoN: A database of gene co-expression network for tea plant (Camellia sinensis), BMC genomics, 2020.

Minor Comments

1. Figure legends should not be repetitions of the methods. Please clearly explain in the legend what is shown in the plot, specially in the supplemental figures Fig. S1-3. You should not expect a general reader to already know the outputs of TMHMM Server. What do probabilities above 1 mean? What are the red peaks?

>Response: We completely agree with the reviewer and we thank reviewer for the comment. The plots of figure S1-S3 show the posterior probabilities of inside/outside/TM helix. The plot is obtained by calculating the total probability that a residue sits in helix, inside, or outside summed over all possible paths through the model. The red peaks represent the possible transmembrane helices present. If the red peak crosses the transmembrane line (represented in pink colour), then it’s calculated as a transmembrane helix. The required modification has been made and the legends are revised for the Supplementary figures as suggested by the reviewer.

2. The tradition set by the original creators of GO is to refer to them as ‘ontologies’ and ‘terms’, rather than ‘groups’ and ‘subgroups’, respectively. Please rephrase to stick with the convention.

>Response: We thank the reviewer for this comment. The required correction has been made in the revised manuscript under the GO ontology analysis and functional interaction network of tea MAPKKKs” in the “RESULTS” section (Page 18). 

3. In the abstract, the sentence “..on the basis of orthologous genes in Arabidopsis, functional interaction was carried out in C. sinensis.” is suggestive of wet-lab experiments. Please rephrase to clearly reflect network analysis of one of the 59 genes the story revolves around.

>Response: The suggested modification has been and marked in the revised manuscript in the “ABSTRACT” section with a line beginning with “Also, on the basis of orthologous…….” (Page 2). 

4. Page 19: I believe that the lines between “Similarity search programs like BLAST…...tend to possess similar functions” are to provide a logical reasoning for your approach of using Arabidopsis network in STRING. If this is the case, please rephrase these lines to reflect the exact logic. It’s somewhat confusing in its current form.

>Response: We thank the reviewer for suggesting the above modification. The required change has been made and marked in the revised manuscript in the “GO ontology analysis and functional interaction network of tea MAPKKKs” section with a line beginning with “All the identified tea MAPKKKs were searched in the TPIA……….” (Page 19).

---

## [Decision Letter · Decision Letter 2]

4 Oct 2021

In-silico genome wide analysis of Mitogen Activated Protein Kinase Kinase Kinase gene family in C. sinensis

PONE-D-21-05595R2

Dear Dr. Mishra,

We’re pleased to inform you that your manuscript has been judged scientifically suitable for publication and will be formally accepted for publication once it meets all outstanding technical requirements.

Kind regards,

Ramegowda Venkategowda, PhD

Academic Editor

PLOS ONE

Additional Editor Comments (optional):

Reviewers' comments:

Reviewer's Responses to Questions

**Comments to the Author**

1. If the authors have adequately addressed your comments raised in a previous round of review and you feel that this manuscript is now acceptable for publication, you may indicate that here to bypass the “Comments to the Author” section, enter your conflict of interest statement in the “Confidential to Editor” section, and submit your "Accept" recommendation.

Reviewer #3: All comments have been addressed

2. Is the manuscript technically sound, and do the data support the conclusions?

Reviewer #3: Yes

3. Has the statistical analysis been performed appropriately and rigorously? 

Reviewer #3: Yes

4. Have the authors made all data underlying the findings in their manuscript fully available?

Reviewer #3: (No Response)

5. Is the manuscript presented in an intelligible fashion and written in standard English?

Reviewer #3: Yes

6. Review Comments to the Author

Reviewer #3: (No Response)

7. PLOS authors have the option to publish the peer review history of their article (what does this mean?). If published, this will include your full peer review and any attached files.

Reviewer #3: No

---

## [Editor Report · Acceptance letter]

25 Oct 2021

PONE-D-21-05595R2 

*In-silico* genome wide analysis of Mitogen Activated Protein Kinase Kinase Kinase gene family in *C. sinensis*

Dear Dr. Mishra:

I'm pleased to inform you that your manuscript has been deemed suitable for publication in PLOS ONE. Congratulations! Your manuscript is now with our production department. 

Kind regards, 

on behalf of

Dr. Ramegowda Venkategowda 

Academic Editor

PLOS ONE